Subject Area:
cognition/neuroscience

Keywords:
autism spectrum disorders, intellectual disability, SYNGAP1, fragile X mental retardation protein, MECP2, NEUROLIGIN

Author for correspondence:
James P. Clement
e-mail: clement@jncasr.ac.in

†These authors contributed equally to this study.

# Understanding intellectual disability and autism spectrum disorders from common mouse models: synapses to behaviour

Vijaya Verma†, Abhik Paul†, Anjali Amrapali Vishwanath†, Bhupesh Vaidya† and James P. Clement

Neuroscience Unit, Jawaharlal Nehru Centre for Advanced Scientific Research, Jakkur, Bengaluru 560 064, Karnataka, India

JPC, 0000-0001-7625-6430

Normal brain development is highly dependent on the timely coordinated actions of genetic and environmental processes, and an aberration can lead to neurodevelopmental disorders (NDDs). Intellectual disability (ID) and autism spectrum disorders (ASDs) are a group of co-occurring NDDs that affect between 3% and 5% of the world population, thus presenting a great challenge to society. This problem calls for the need to understand the pathobiology of these disorders and to design new therapeutic strategies. One approach towards this has been the development of multiple analogous mouse models. This review discusses studies conducted in the mouse models of five major monogenic causes of ID and ASDs: *Fmr1*, *Syngap1*, *Mecp2*, *Shank2/3* and *Neuroligins/Neurnexins*. These studies reveal that, despite having a diverse molecular origin, the effects of these mutations converge onto similar or related aetiological pathways, consequently giving rise to the typical phenotype of cognitive, social and emotional deficits that are characteristic of ID and ASDs. This convergence, therefore, highlights common pathological nodes that can be targeted for therapy. Other than conventional therapeutic strategies such as non-pharmacological corrective methods and symptomatic alleviation, multiple studies in mouse models have successfully proved the possibility of pharmacological and genetic therapy enabling functional recovery.

## 1. Introduction

The human brain is a complex organ with a wide array of functions. An adult brain has approximately 86 billion neurons and 85 billion non-neuronal cells [1]. Synchronized activity among neuronal and non-neuronal cells enables us to perform from mundane yet straightforward tasks to an overly complicated range of activities. Development of the human brain is a tightly regulated process. Any change can lead to precarious and detrimental developmental deficits such as neurodevelopmental disorders (NDDs). To name a few, NDDs include, but are not limited to, autism spectrum disorder (ASD), intellectual disability (ID) and attention deficit hyperactivity disorder (ADHD), which affect 3–4% of the world's population [2]. On average, NDD is diagnosed when a child is six months to 1 year old in the absence of well-defined biomarkers as the child is not fulfilling developmental milestones. The *Diagnostic and Statistical Manual of Mental Disorders* (DSM V), published by the American Psychiatric Association in 2013, has suggested the following for the diagnosis of ID: children often present with difficulty in learning and memory, and exhibit deficits in self-care and social behaviour [3].

Studies over the past few decades have shown that there is a strong genetic correlation between specific genes encoding protein synthesis that regulates synaptic function and ASD/ID [4]. Mutations in such genes, along with gene–gene and gene–environmental interactions, are responsible for ASDs and ID.

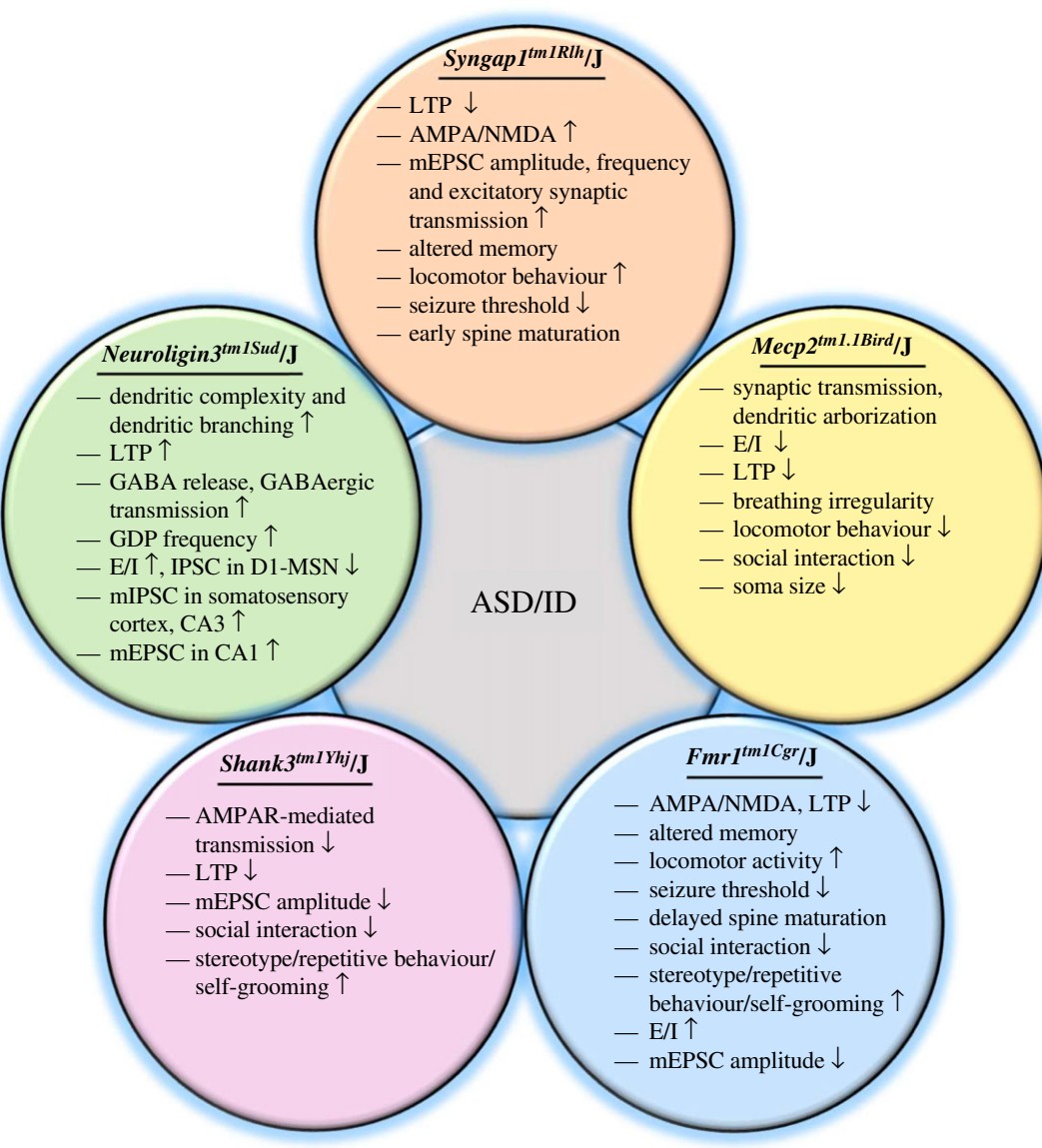

**Figure 1.** Common pathophysiological features observed in genes implicated in ID/ASDs. Diagram illustrating morphological, synaptic and circuit properties of neurons along with behavioural alterations observed in different key transgenic mouse models. As mentioned in the main discussion, although the mutations were observed in different genes that are implicated in ASDs/ID, there are many common features found in these mutations. Therefore, it is imperative to understand the mechanism of these mutations about neuronal function before prescribing therapeutics to patients with any of these mutations. E/I, excitation—inhibition; GABA, gamma-aminobutyric acid; GDP, giant depolarization potential; D1-MSN, D1 receptors in medium spiny neurons (MSN); AMPAR, α-amino-3-hydroxy-5-methyl-4-isoxazolepropionic acid receptor; mEPSC, miniature excitatory postsynaptic current; mIPSC; miniature inhibitory postsynaptic current.

The resulting pathophysiological mechanisms of many of these overlap with those of ID and ASD, such as *SYNGAP1*, *FMRP*, *MECP2*, *NEUROLIGINS*/*NEURONEXINS* and *SHANK2*/3. Apart from the overlapping pathophysiology, patients with ASD and ID have some comorbid conditions which include schizophrenia, allergic disorders, food sensitivity and autoimmune disorders to name a few [5–8].

While the distribution of specific proteins such as FMRP is widespread, the SYNGAP1 protein is expressed only in the brain, not in any other organ [9,10]. The variation in the tissue, cellular and subcellular localization and expression of these genes implicated in ID and ASDs may be one of the reasons for the type of phenotypes observed in ID/ASD individuals. Genes such as *FMRP* present with the syndromic type of ID while others such as *SYNGAP1*, *MECP2* and *SHANK3* are mostly linked to the non-syndromic type [11,12]. The environmental factors which can lead to ID and ASDs include the use of certain drugs during pregnancy like valproate and alcohol as well as infections, exposure to heavy metals, such as lead and mercury, and malnutrition [13–16]. Although a complex interplay of environmental and genetic factors are known to have a role in the pathophysiology of ID/ASDs, most cases are idiopathic [17].

Precise control of synapse formation and development is essential for correct brain development and function. Abnormalities, if any, can lead to various biochemical and behavioural deficits. In this review, we have discussed the convergence of the pathophysiological hallmarks and phenotypic characteristics with emphasis on the changes in the synaptic morphology (figure 1). For instance, alterations in the α-amino-3-hydroxy-5-methyl-4-isoxazolepropionic acid receptor/*N*-methyl-ᴅ-aspartate receptor (AMPA/NMDA) ratio, induction and maintenance of long-term potentiation (LTP) and long-term depression (LTD), and changes in the basal synaptic transmission are some of the common characteristics associated with ID and ASDs of different genetic backgrounds, which are discussed in detail in this review [18]. The current report further highlights the

importance of the critical period of development and its alteration due to pathogenic mutations [19].

Although the importance of the proper function of neurons during a critical period of development is known, the role of non-neuronal cells such as astrocytes has been barely studied, but this has been changing in recent years. It has been observed that the timing of astrocyte maturation coincides well with the formation of excitatory synapses in the brain. Secretion of molecules such as glypicans by astrocytes helps in the conversion of silent to functional synapses, thereby facilitating the insertion of AMPA receptors [20]. Although the expression of synaptic proteins such as FMRP and NEUROLIGIN has been studied in astrocytes, a complete understanding of their underlying roles in ID and ASDs is still under investigation [21]. Here, we give an overview of the importance of astrocytes during the critical period of development, and how alteration in these impacts neuronal function.

To further our understanding of the physiological relevance of mutations implicated in ID and ASDs, the use of transgenic animal models has provided insights into different pathophysiological aspects of ID and ASDs. These transgenic mice models had been validated at various levels to ensure their efficacy in the replication of not only the pathophysiology but also the behavioural phenotypes [22]. Although there are limitations, they have been useful in understanding the novel mechanisms contributing to the progression and development of these disorders and in finding novel therapeutic targets.

As with most neurological disorders, the exact prevalence and epidemiology of NDDs are not entirely known yet. Meta-analysis studies, which make use of statistical procedures to analyse data from already existing reports, have highlighted the procedural limitations and underreporting cases from several parts of the world [23,24]. However, current studies have shown the prevalence to be higher in low- and middle-income countries, which can be attributed to the lack of essential diagnostic and management resources in these geographical locations [23,24]. Considering that NDDs require early diagnosis, an exhaustive study done in children from birth until 12 years of age found the prevalence to be as high as one in six children in the USA. Moreover, in the same study, the incidence was higher among the males than among females [25].

Nevertheless, according to a recent report by the World Health Organization, about one in every 160 individuals has an ASD/ID [26]. The numbers are remarkably alarming as they are only expected to become worse in the absence of any effective therapeutic strategy. With more laboratories now working on NDDs, insights into possible new therapeutic targets and their mechanisms may aid finding mitigation strategies in the future.

Drug repurposing is one of the impending fields; this involves finding a new indication for an already approved drug [27]. Drug development is an expensive and time-consuming process that could be shortened with the help of this approach. In this review, we have highlighted studies done on drugs with a known indication in some other diseases. Targeting them at the preclinical and clinical stages may prove to be a useful strategy in search of new medications for ID and ASDs.

Although there are several unexplored mechanisms associated with ID/ASDs, the involvement of synaptic function and plasticity in ID and ASDs is well characterized [28].

Technological advancement to measure neuronal activity (electrophysiology and deep-brain imaging) and understanding of the mechanisms which modulate synaptic plasticity may aid in further expanding our knowledge to decipher the pathophysiology of ID and ASDs. In this review, we will discuss the implications of monogenic mutations on the physiological, molecular and biochemical, and morphological aspects of neuronal and non-neuronal development using different mouse models studied over the past few decades.

## 2. Aetiology/causes of intellectual disability

The causes of ID and ASDs are diverse and involve a range of genetic and environmental factors [29]. Although the nature of the cause for about 60% of all known cases of ID and ASDs remains unknown [30,31], studies where the cause is known have demonstrated that aberrations leading to ID/ASDs mainly occur during the developmental time period, but have a lifelong effect, including in adulthood. The vulnerable range of time includes the pre-, peri- and post-natal stages of development [32]. Environmental stress factors such as poor nutrition, hygiene, infection, familial instability and socio-economic causes may affect brain development, contributing to ID/ASDs [33,34]. For example, it is now believed that oxidative stress as a result of environmental stressors such as heavy metals affects sulfur metabolism, which leads to alteration of the epigenetic mechanisms of gene expression. Hence, the complex interplay of genetic and environmental factors has a crucial role in the pathology of ASDs/ID [17].

Other than environmental factors, about a quarter to half of the identified causes of ID are the result of mutations in genes [32]. These mutations can be either inherited or acquired genetic defects due to metabolic genetic defects as observed in cases such as phenylketonuria and Tay–Sachs disease that can disrupt healthy brain development [33]. These genetic changes can occur on various scales based on size (and thereby ease of detection)—the largest being microscopically visible aberrations at the chromosomal level and 15% of all cases of ID were attributed to these chromosomal defects [30,35–38]. This review will provide a brief overview of the current understanding of ID/ASDs with regard to the critical period of development, molecular and biochemical signalling, and electrophysiological aberrations concerning monogenic mutations.

## 3. The critical period of plasticity

A remarkable property of the brain is its capability to undergo changes based on experiences (stimuli) through a process called synaptic plasticity, particularly during the early stages of development [39]. Synaptic plasticity is a biological process in which a stimulus or experience induces synaptic activity that results in changes in synaptic strength and contributes to learning and memory. It includes the formation, storage and retention of sensory memories, and occurs throughout life starting from birth and continues into adulthood to an extent. This process allows us to do everyday activities such as performing mathematical calculations, decision making and more complex multi-tasking activities, which are coordinated by proper neuronal connections in the brain that act as the central processing unit of the body. However, at the time

royalsocietypublishing.org/journal/rsob   Open Biol. 9: 180265

of birth, there are fewer synaptic connections, but, as the brain matures, there is a massive surge in the number of neuronal connections that enables a child to learn tasks rapidly [40]. During this stage of development, the synaptic connections are highly vulnerable to changes in environmental stimuli. However, at the end of adolescence, only active connections (upon repeated or stronger stimulation) will be strengthened, and less active connections (upon weaker or not repeated stimulation) will be eliminated [41]. For example, a child may learn a new language in a few months to a year, but it might take months to a few years for an adult to master the same language. [42]. These observations postulated the phenomenon of a critical period of neuronal development. Different experiences during early post-natal life determine the process of formation, maturation and elimination of neuronal connections in the brain. Windows of heightened neuronal plasticity during brain development are termed as critical or sensitive periods, which predominantly occur in early life, and, hence, learning any new paradigms is relatively easy and quick in childhood, although it continues at a reduced level in adulthood. A critical period is a time when environmental or sensory input is required for the proper development of particular neuronal connections in different regions of the brain [43,44]. If these connections are unstimulated, the brain function served by that circuit will be permanently compromised [43,44]. For example, the critical period for the visual cortex begins soon after a baby is born in humans or soon after eye opening in animals such as rats or mice. Similarly, the onset of hearing triggers a critical period of development of the auditory cortex in humans. Perturbation in neuronal connections of these regions before the end of the critical period might permanently compromise their function. In the last decade or so, several laboratories have used visual, auditory and thalamocortical regions of the brain as models to understand the importance of the critical period in the plasticity and development of an individual [41,45–47].

An important contribution to using the visual cortex as a model came from studies by Hubel and Wiesel in 1962; they emphasized the importance of the stimulus-dependent response by a population of neurons. They observed that, for cortical cells, the most effective stimulus configurations dictated by the spatial arrangement of excitatory and inhibitory regions were long narrow rectangles of light (slits), straight line borders between areas of different brightness (edges) and dark rectangular bars against a light background. To attain the maximum response, the shape, area and orientation of these stimuli were critical [48]. Recordings from different stages of development in normal kittens and those with monocular dominance (MD), where one eyelid was sutured during the early time window of development, concluded that kittens with MD had almost irreversible changes in the functional properties of the visual cortex area V1, suggesting the importance of sensory-dependent activation of neurons during the critical period of development and its relevance to neuronal connections [41]. These studies have demonstrated the importance of the critical period in the visual cortex and the modulation of neuronal plasticity based on experience/sensory information. For a better understanding of the molecular mechanisms underlying the critical period of development and plasticity in the primary visual cortex, studies using various animal models are discussed extensively in [41,48–53], and similar information can be obtained for thalamocortical studies from [54–57].

Studies have shown that changes at the synaptic receptor number and subunit expression of both excitatory and inhibitory neurons signal the opening or closure of a critical period of neuronal development [58]. For example, based on these studies, NMDARs are considered as one of the molecular determinants of the critical period of plasticity as NMDAR-mediated synaptic transmission is developmentally regulated, and their expression can modify neuronal activity [58–61]. In the visual cortex, the percentage of total NMDAR-mediated current can be described by a slow exponential decay between the first and fifth post-natal week. Dark rearing of pups delays the developmental shortening of NMDAR-mediated currents, suggesting that the change in the 2A/2B ratio requires stimulus-based experiences in the visual cortex, and impairs the closure of the critical period. Pups reared in the dark displayed longer duration of NMDAR-mediated currents similar to younger animals bred in normal conditions, suggesting an altered critical period of development [62]. Apart from the visual cortex, various laboratories have studied the critical period of development in the thalamocortical region of the brain. By focusing on the NR2B to NR2A switch, which is developmentally regulated, NMDAR-mediated currents were recorded in layer IV of rat somatosensory cortex and found to be decayed more rapidly in PND7 than in PND3, which further suggests NMDAR-mediated, mainly subunit switch, modulation of the critical period [63–65].

Another key factor that regulates the critical period of plasticity during development is the function of gamma-aminobutyric acid (GABA) in the neuron. The function of GABA is carried out by two $Cl^-$ cotransporters, NKCC1 and KCC2, and its expression varies during the early stages of development [66]. Several studies using the gene disruption method to modulate the function of GABA and chloride concentration suggested the possibility to target NKCC1 and KCC2 for the rewiring of neuronal connections in the brain post-critical period of development, and, hence, a potential target for therapeutics [67–69]. In various NDDs, such as ID, ASDs and schizophrenia, one of the common features observed is an imbalance of the excitation–inhibition (E/I) ratio, which can serve as a key factor in understanding the major cause of these disorders, and how manipulating the critical period, particularly targeting GABA, would help to resolve the associated defects [70,71]. It has been known for decades that maturation of the inhibitory cortical circuits in the brain parallels the opening of the critical period [72–76]. With the help of glutamate decarboxylase (GAD)–knockout (KO) mice (GAD is a GABA-synthesizing enzyme), it has been demonstrated how inhibitory signalling regulates the critical period. For example, GAD-65 KO mice have impaired ocular dominance shift when one eye is deprived of vision [72]; visions was restored by the application of diazepam (benzodiazepine), which enhances GABA activity, within the critical period window by enhancing GABAR activity. These experiments suggest that ocular dominance plasticity is regulated by inhibitory neuronal signalling in the brain, apart from the excitatory signalling as discussed earlier. GAD-65 KO mice had shown reduced NR2A levels and slower NMDAR-mediated currents in the visual cortex, which plays an important role in the critical period of plasticity [74]. Other studies have focused on the ectopic expression of BDNF in mice, which is also known to regulate GABAergic inhibitory interneurons and related synaptic strength and has been shown to induce early opening and premature

royalsocietypublishing.org/journal/rsob    Open Biol. **9**: 180265

closing of the critical period of plasticity [73,77]. These studies suggest that there are different molecular determinants of the critical period of plasticity that are developmentally regulated. Here, we have briefly discussed the role of different proteins, especially GABAR, which are developmentally regulated and, in turn, modulate the critical period of plasticity. Thus, altering GABAR-mediated functions could be a potential therapeutic approach in the rewiring of synaptic connections and rescuing the pathophysiology of NDDs, particularly after a critical period of development, which is one of the major challenges faced by many neuroscientists.

# 4. Animal models of intellectual disability

The use of animal models to understand disease pathogenesis and to design treatment strategies has been long been a practice in biology and related disciplines. Despite the substantial ethical debate surrounding the use of animals in research, and the argument of interspecific variations, data from animal models remain the most positive attribute of biomedical research [78]. With technological advancements and the advent of transgenic mice, the ability to study more complex biological problems became feasible as a result of genetic modifications [79]. More recently, the use of Cre-Lox technology to generate tissue-specific KOs and other methods of modulating gene transcription *in vivo* has further contributed to our understanding of the mechanisms of human disease [80,81].

Although it is not possible to mimic all aspects of a disorder or disease in any one animal model, a suitable animal model should be able to replicate the clinical hallmarks of the disease with the paramount degree of robustness. Therefore, for a given condition, a range of animal models are usually defined, characterized and then validated. A good animal model should meet the following criteria: internal validity, external validity, construct validity, face validity and predictive validity. Internal validity refers to the reliability and reproducibility of the model with regard to consistency in the experimental measurements. Face validity describes the degree of similarity between the symptoms shown in human populations and those expressed in the animal model. Predictive validity involves the extrapolation of a particular experimental manipulation done in a species or a specific situation to the other species and situation. Construct validity refers to the degree of similarity in the mechanisms underlying the behavioural similarities between the animal model and those seen in patient populations. External validity involves the generalizability of the results obtained in the animal studies in relation to the general population [22,82].

Different reviews have discussed the commonly available and used animal models for ID [82–84]. Some of the widely used mouse models that replicate the pathophysiology of ID are listed in table 1 along with the morphological, biochemical and behavioural alterations associated with them. Nevertheless, citing the failure of pharmacological agents in those clinical trials which otherwise showed efficacy in the preclinical studies, it becomes clear that the existing models have their limitations [79]. Hence, there is a need to look for new animal models of ID/ASDs that display different mutations and not only reproduce the clinical features of a disease but also guarantee a higher degree of translational success.

# 5. Gene mutation in intellectual disability and autistic spectrum disorders

## 5.1. SYNGAP1

SYNGAP1 is a 135 kDa protein that was shown by Chen *et al.* [116] and Kim *et al.* [10] to be one of the targets of phosphorylation by $Ca^{2+}$/calmodulin-dependent protein kinase II (CaMKII) in the post-synaptic density (PSD) in the rat brain. It has several different isoforms which arise as a result of alternative splicing owing to different start sites. Though the existence of SYNGAP1 isoforms was first identified in 1998, detailed characterization and analysis were done much later [82,116,117]. The functions of different isoforms and physiological functions of *Syngap1*, particularly in regard to the *Syngap1* mutation and its implications in ID and ASDs, are extensively discussed in another comprehensive review by Jeyabalan & Clement [82].

### 5.1.1. Behavioural changes associated with *Syngap1* mutation

$Syngap1^{+/-}$ was found to be associated with several behavioural abnormalities, including cognitive and learning deficits, reduced seizure threshold, hyperactivity and increased locomotion [118,119]. Behavioural tests such as the Morris water maze, radial arm maze, spontaneous alternation test and the Y-maze novel arm test have been used to compare learning- and memory-related impairments in $Syngap1^{+/-}$ mice [87,120]. $Syngap1^{+/-}$ mice showed a significant decline in working memory; however, the performance was comparable to wild-type (WT) animals in the reference memory tasks, suggesting only specific memories are impaired [121]. These mice further presented with deficits in the remote memory when tested on the contextual fear learning procedure [119]. These findings have been verified physiologically by Clement *et al.* [88] by measuring basal synaptic transmission from the dentate gyrus and relating it to learning and memory deficits observed in these mice and patients.

Apart from cognitive decline, $Syngap1^{+/-}$ mice also exhibited stereotypic behaviour, hyperactivity and reduced anxiety-like behaviour, which was estimated experimentally using an open field test and elevated plus maze [85,121]. These mice spent more time in the open arms of the elevated plus maze than the WT animals, thereby displaying enhanced overall locomotion [85,119,121]. These behavioural abnormalities suggest impairment in learning and memory. To further validate these findings related to the *Syngap1* heterozygous mutation, a study made use of the cliff avoidance test in which latency to jump off the cliff was higher in the case of the $Syngap1^{+/-}$ mice than in the normal WT animals, suggesting the lack of competency to judge the depth [122]. Similarly, patients with the *SYNGAP1* mutation exhibited a non-syndromic form of ID that had been linked to a decline in sociability, and this was reinforced with the help of sociability tests done in transgenic mice [85].

Studies from patients with the *SYNGAP1* heterozygous mutation have shown that approximately 80% of patients have epileptic seizures [82,123,124]. Indeed, electroencephalogram monitoring combined with video monitoring assays in patients showed spontaneous abnormal cortical activity in patients with the *Syngap1* heterozygous mutation

**Table 1.** Tabulation of behavioural, synaptic and biochemical alterations used in transgenic mouse models of different ID/ASD-related genes (*Fmr1, Syngap1, Shank, Neuroligin3 and Mecp2*). LTD, long-term depression; Pv, parvalbumin; N/D, not defined or determined; EPSC, excitatory postsynaptic current; E/I, excitation–inhibitory; GTP, guanosine-5'-triphosphate.

| genetic modification | behavioural changes | changes in synaptic morphology and function | biochemical alterations | references |
|---|---|---|---|---|
| *Syngap1* mutation | | | | |
| exon 7/8 in *Syngap1*$^{+/-}$ mice B6.129-*Syngap1*$^{tm1Rlh}$/J **MGI:** 3822367 | stereotypic behaviour, anxiety↓, memory deficits and social interaction↓ | LTP↓, AMPA/NMDA↑ | N/D | [85,86] |
| exon 4–9 in *Syngap1*$^{+/-}$ mice, *Syngap1*$^{tm1.1Mabk}$ **MGI:** 3511175 | | mEPSPs↑, early maturation of the spines | altered clustering of PSD-95 protein and their movement into the spine head, dysregulation of Ras, activation of the Rho family of GTP-binding proteins and phosphatidylinositol-3-kinase | [87] |
| exon 5/6 and 7/8 in *Syngap1*$^{+/-}$ mice B6.129S2 *Syngap1*$^{tm2Geno}$/RumbJ **MGI:** 5796355 STOCK *Syngap1*$^{tm1.1Geno}$/RumbJ **MGI:** 5796354 | seizure threshold↓, altered context discrimination behaviour, locomotor activity↑ | early spine maturation, AMPA/NMDA↑, LTP↓ | N/D | [88] |
| *Fmr1* mutations | | | | |
| *Fmr1*-KO (neomycin cassette inserted into exon 5) B6.129P2-*Fmr1*$^{tm1Cgr}$/J **MGI:** 2162650 | cognition↓ and activity↑ seizure threshold↓, sensitivity to sensory stimuli, anxiety↑ social interaction↓ | spine density, immature thin, elongated spines↑ | group I mGluR-mediated LTD↑ local protein synthesis↑ | [89,90] |
| *Fmr1*-KO2 (germline ablation of promoter and first coding exon) *Fmr1*$^{tm1.1Cidz}$ **MGI:** 3808885 | hyperactivity altered emotional processing memory deficits hypersensory response ultrasonic vocalizations↓ | spine heads↓ and wider spine necks | AMPA/NMDA↓ NMDAR-mediated LTP↓ | [91–93] |
| *Fmr1*-CKO (promoter and first coding exon are floxed, can be removed with conditional cre-expression) *Fmr1*$^{tm1Cidz}$ **MGI:** 3603442 | hippocampus-dependent learning deficits cerebellar eyelid conditioning↓ | immature spine number↑ | LTD↑ | [94,95] |
| *Mecp2* mutation | | | | |
| *Mecp2* KO B6.129P2(C)-*Mecp2*$^{tm1.1Bird}$/J **MGI:** 2165230 | movement↓, improper gait, hind limb clasping, respiratory disorder | number of dopaminergic neurons↓, soma size↓, precocious opening of critical period in visual cortex and accelerated maturation of GABAergic PV(+) neurons | deficit in GABA and glutamate synthesis pathway, spatio-temporal alteration of NMDAR expression, alteration in activity-dependent global chromatin dynamics | [96–102] |

(Continued.)

**Table 1.** (Continued.)

| genetic modification | behavioural changes | changes in synaptic morphology and function | biochemical alterations | references |
|---|---|---|---|---|
| *Mecp2*-CKO *TH-Cre, Mecp2* ^flox B6.129P2-*Mecp2*^tm1Bird/J **MGI:** 3702570 | total distance and vertical activity in open field↓, performance in dowel walking test↓ | dopamine, norepinephrine, serotonin release↓ | expression of *TH* and *Tph2*↓ | [103] |
| *MeCP2* KI FVB-Tg(MECP2)1Hzo/J **MGI:** 3817212 | homozygous animals show tremors, gait ataxia↑ heterozygous animals show rescue from RTT-like symptoms | neuronal cell number and brain size is rescued to wild-type littermates | N/D | [104] |
| *MeCP2*^R168X point mutation; STOCK *Mecp2*^tm1.1Jtc/SchvJ **MGI:** 5568127 | stereotyped behaviour↑, hypoactivity, breathing problems | N/D | no change in the *Ube3A* mRNA level | [105] |
| *Shank3* mutation | | | | |
| *Shank3* ^e4−9 homozygous B6.129S7-Shank3^tm1Yhj/J **MGI:** 5295948 | social interaction↓, repetitive behaviour↑, impaired memory | activity-dependent redistribution of GluA1 AMPAR↓, thin long dendritic spines↑, LTP↓ | GKAP, PSD95, Homer protein level↓ | [106] |
| *Shank3* ^e4−9 heterozygous B6[36]-Shank3^tm1.2Bux/J **MGI:** 5317118 | social behaviour, social sniffing, ultrasonic vocalization↓ | mEPSC, basal neurotransmission, LTP↓ | AMPAR expression↓ | [107] |
| *Shank3B*^−/− B6.129-Shank3^tm2Gfng/J **MGI:** 5444207 | repetitive grooming↑, social interaction↓ | complexity↑ of dendritic length, dendritic arborization↑ and ↓surface area of MSN, caudate volume, cortico-striatal synaptic transmission, mEPSC frequency in MSN | SAPAP3, PSD93, Homer, NR2B, GluA2, NR2A expression↓ | [108] |
| Neuroligin | | | | |
| *Neuroligin3* R451C knockin mice B6.129-Nlgn3^tm1Sud/J **MGI:** 3820515 | rotarod-mediated motor behaviour↑ | dendritic complexity and dendritic branching in hippocampus↑. mEPSC in CA1, mIPSC in somatosensory cortex, LTP↑. mIPSC in CA3, GABA release, GDP frequency↑. GABAergic synaptic transmission↑ and IPSC amplitude↓ in barrel cortex and hippocampus. IPSC and E/I ratio in D1-MSN | NLGN3 protein misfolding and trafficking defects, NLGN3 expression was 90%↓. Alteration of NMDAR subunit composition and expression of NMDAR subunit 2B↑. IPSC amplitude and success rate at the same synapse failed to respond to AM251 (CB1 receptor antagonist) | [109–114] |
| *Neuroligin3* R704C knockin mice STOCK Nlgn3^tm3.1Sud/J **MGI:** 5437466 | | AMPAR-mediated synaptic response↓ and unaltered NMADR or GABAR-mediated response. Unaltered NMDAR-mediated LTP, EPSC frequency↓ and NMDA/AMPA, in cultured hippocampal neurons↑ | levels of AMPAR subunits GluA1 and GluA3↑ | [115] |

[119]. Similarly, $Syngap1^{+/-}$ mice exhibited a reduction in the seizure threshold and induction of myoclonic seizures as a result of mutations in the $Syngap1$ gene, which is in agreement with the epileptic seizures observed in patients [88,119]. Additionally, reduced levels of SYNGAP1 in the inhibitory GABAergic neurons exhibited reduced inhibitory synaptic activity and cortical gamma oscillation power and resulted in cognitive and social deficits [125].

A recent report has shown the impairment of sensory processing and touch-related deficits observed in patients as a result of reduced activity within the upper lamina somatosensory cortex (SSC) circuits in $Syngap1^{+/-}$ mice [126]. On the contrary, an earlier finding suggested that $Syngap1$ heterozygous mutation resulted in increased overall excitability of neurons [126]. These studies demonstrate the fact that $Syngap1^{+/-}$ mutations lead to abnormal neuronal activity, thereby causing excitatory and inhibitory imbalance.

In addition, the clinical findings were similar to the behavioural phenotypes observed in mice. For example, patients with $SYNGAP1^{+/-}$ reported delated psychomotor symptoms and developmental delays [127–129]. They also manifested reduced seizure threshold and increased chances of epileptogenesis [124,129–131]. Identification of different splice variants of the $SYNGAP1$ gene in patients would help to understand the implication of this gene mutation in humans and the translational success of studies done in rodents [132,133].

## 5.1.2. Synaptic function and morphology

SYNGAP1 protein has been shown to regulate postsynaptic cytoskeletal changes and AMPA receptor trafficking onto the surface of the postsynaptic membrane [86]. It was reported that $Syngap1^{-/-}$ mice died within a week of their birth, while the survival of $Syngap1^{+/-}$ mice was similar to WT [86]. The latter further showed a reduction in LTP in comparison with the WT littermates, suggesting impaired learning and memory abilities [86]. Non-viability of the $Syngap1^{-/-}$ mice has been attributed to increased apoptosis as a result of CASPASE-3 activation [134]. This could explain the reasons for not identifying any homozygous mutations in human patients, to date, and corroborates well with the preclinical mouse model, suggesting its impact on translational research. $Syngap1^{+/-}$ mice, on the other hand, exhibited premature spine maturation, leading to an overall increase in neuronal excitability [87]. Research has also pointed out the involvement of SYNGAP1 in the ACTIN-mediated steady-state regulation of spine morphology, which is necessary for spine maturation [135]. Clement $et\ al.$ [88] confirmed the association of behavioural abnormalities with premature spine maturation in the hippocampus. They observed a higher number of mushroom-shaped spines and a lesser number of stubby spines at the beginning of the second post-natal week in comparison with the WT littermates [88]. This confirmed the role of SYNGAP1 in the control of cytoskeleton rearrangement and spine maturation. A follow-up study went on to suggest the crucial role of SYNGAP1 during the critical period of development [136]. Apart from early spine maturation as previously reported, somatosensory neurons in the $Syngap1^{+/-}$ mice showed adult neuron-like characteristics, including increased arbour complexity, total length and occupational volume [19]. Similar observations were made in another report, which studied the development of layer II/III of the medial prefrontal cortex (mPFC) in $Syngap1^{+/-}$ mice. They found

an elevated AMPA/NMDA ratio during the early stages of development, which was correlated to the prematuration of excitatory synapses in the cortex [136].

Maturation of the spines from filopodia to mushroom-shaped requires un-silencing of synapses mediated by an increase in the insertion of AMPA receptors. This change leads to an increase in the level of basal synaptic transmission, which has already been discussed in detail elsewhere [137]. It was initially demonstrated that the basal synaptic transmission remained unaffected in the adult $Syngap1^{+/-}$ mice. However, a follow-up study by Clement $et\ al.$ [88] showed that AMPAR-mediated currents increased in P14–16, equivalent to that of WT, but otherwise remained unchanged in young (P7–9) and adult mice [88]. This suggested that the alteration in the insertion of AMPA receptors could lead to changes in the excitatory and inhibitory balance and altered the critical period of development which eventually causes various behavioural defects as reported by patients with $SYNGAP1^{+/-}$.

As a part of the NMDAR-mediated signalling pathway, a heterozygous mutation in $Syngap1$ may impair the NMDAR-mediated current. However, it remained unaffected when NMDAR-mediated currents were measured from different stages of development, suggesting a role for $Syngap1$ in synapse formation and function without altering the characteristics of NMDARs [138]. An $in\ vitro$ study performed later went on to show an increase in the amplitude and frequency of the mEPSCs as a result of an increase in the number of AMPA receptors at the post-synapse and, hence, the surge in the AMPAR-mediated current [87]. In a similar study done $in\ vivo$, comparable results were obtained, though only at the P14–16 stages [88]. These studies further suggested that an increase in the AMPA/NMDA ratio at P14–16 in $Syngap1^{+/-}$ mice correlated with an increase in the number of functional synapses in the hippocampus. Experiments involving other areas of the brain such as mPFC have also yielded similar results with an overall increase in glutamatergic activity [119].

As a result of increased basal synaptic transmission and excitability due to increased AMPA receptor insertion at the post-synapse, LTP generation and maintenance were impaired in the adult $Syngap1^{+/-}$ mice [86,119,138]. On the other hand, the effect of $Syngap1^{+/-}$ on LTD induction has been studied less extensively. Acute application of NMDA had been widely used to induce LTD in acute brain slices [139]. The same protocol when used in $Syngap1^{+/-}$ mice yielded poor maintenance of the LTD in comparison with the WT control animals where stable LTD was maintained throughout the experiment [135]. Nevertheless, the paired-pulse protocol used elsewhere did not show significant differences in LTD induction in $Syngap1^{+/-}$ mice, which suggests that the release probability was not affected by SYNGAP1 and it was an altered function of post-synapses [86]. To study the role mGluR-mediated LTD, Barnes $et\ al.$ [18] stimulated group I mGluRs with dihydroxyphenylglycine (DHPG) in the hippocampus and showed mGluR-LTD was significantly increased independent of protein synthesis in $Syngap1^{+/-}$ mice at PND 25–32. Our unpublished data, at the time of writing this review, further confirmed that increased mGluR-LTD is persistent in adulthood. However, the mechanisms of how SYNGAP1 regulates mGluR-LTD are yet to be elucidated. One of the possible mechanisms proposed by Barnes $et\ al.$ suggests the

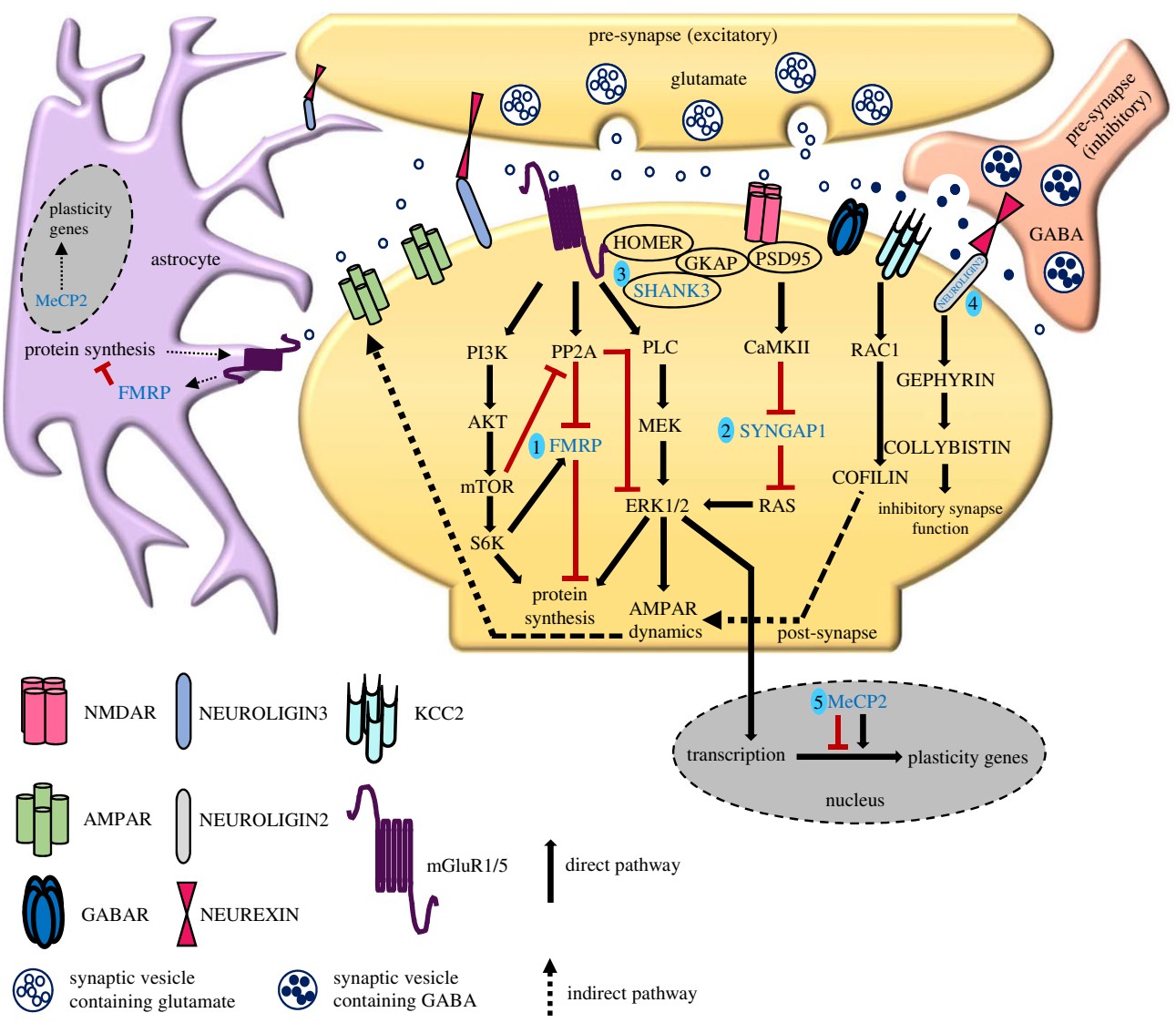

**Figure 2.** Illustration depicting primary signalling mechanisms of different neuronal and astrocytic proteins encoded by genes implicated in ID/ASDs. Activation of the pre-synapse leads to neurotransmitter release. The neurotransmitter binds to the corresponding receptors; these allow the influx of divalent ions that trigger several downstream signalling cascades. (1) Activation of group I mGluRs leads to dephosphorylation of FMRP by PP2A. This dephosphorylation displaces FMRP from mRNA promoting their translation. Simultaneously, PP2A inhibits extracellular signal-regulated kinase (ERK)-mediated protein synthesis. On a slower time scale, mGluR activation stimulates the mTOR pathway, which, consequently, re-phosphorylates FMRP and inhibits mRNA translation. (2) NMDAR activation leads to phosphorylation of CaMKII, which in turn phosphorylates SYNGAP1 and traffics AMPARs to the postsynaptic membrane via ERK. (3) SHANK3 is expressed downstream of group I mGluRs and regulates signalling via HOMER, which might also regulate NMDAR-mediated signalling. In addition, SHANK3 interacts with NMDAR via GKAP and PSD95, thereby regulating synaptic plasticity. (4) NEUROLIGIN3 interacts directly with NEUREXINS and maintains the stability of the excitatory synapse, whereas NEUROLIGIN2 regulates inhibitory synapse function via GEPHYRIN and COLLYBISTIN. (5) MeCP2 regulates expression of different plasticity-related genes in neurons as well as in astrocytes.

presence of convergence in the biochemical signalling pathway downstream of mGluR- and NMDAR-mediated signalling proteins, although this needs to be further investigated in detail.

### 5.1.3. Biochemical pathways and their alteration in the *SYNGAP1* haploinsufficiency

SYNGAP1 is a postsynaptic protein that is downstream of NMDA receptors and the postsynaptic scaffolding protein PSD-95 [10]. It negatively regulates RAS-GTPase activity, thereby regulating the insertion of AMPARs at the post-synapses [10]. Furthermore, its activity is regulated by the phosphorylation of CaMKII [10]. Based on these initial findings, the possible phosphorylation sites of SYNGAP1 were recognized, and the levels of phosphorylation increased activation of NMDARs [140]. Studies performed using

*Syngap1*$^{+/-}$ mice to identify the signalling cascades downstream of SYNGAP1 demonstrated that it is a negative regulator of extracellular signal-regulated kinase/mitogen-activated protein kinase (ERK/MAPK) signalling and a positive modulator of the p38–MAPK signalling pathway, thereby regulating activity-induced synaptic plasticity (figure 2 compares the signalling mechanisms impaired in different mutations in ID that are discussed in this review) [138,141]. Additionally, the activity of proteins such as p21-activated kinase [27], RAC and p-Cofilin, which are regulated by the SYNGAP1, were also elevated in *Syngap1*$^{+/-}$ mice under the basal conditions [135]. Thus, SYNGAP1 regulates spine morphology and function by modulating cytoskeletal dynamics. It was clear from these studies that the signalling pathways downstream of NMDARs were impaired in *Syngap1*$^{+/-}$, causing various sensory, cognitive and social deficits as observed in patients.

## 5.1.4. The critical period of plasticity

*Syngap1* is shown to have a developmentally regulated expression in mice wherein the expression peaks at PND14 and stabilizes subsequently. In *Syngap1*$^{+/-}$, at PND14–16, the SYNGAP1 expression was half of that of WT. Electrophysiological studies in acute hippocampal brain slices have shown an increased AMPA/NMDA ratio which was further corroborated by an increased number of mushroom-shaped spines at PND14, increased basal synaptic transmission and increased mEPSC and mIPSC (increase in amplitude and frequency, which suggests that the number of functional AMPAR/GABAR, respectively, in the post-synapse) at PND14 in *Syngap1*$^{+/-}$ when compared with WT (matures at PND21) [88]. Overall, this study suggests that early maturation of dendritic spines, which indirectly illustrates the altered critical period of plasticity and development in *Syngap1*$^{+/-}$. Clement *et al.* in 2012 [88], using the conditional Cre-lox system to create haploinsufficiency of SYNGAP1, measured AMPA/NMDA in neonatal and young adult mice. Surprisingly, they observed that the AMPA/NMDA ratio in *Syngap1*$^{+/-}$ young adults was the same as WT adults, which contradicts increased AMPA/NMDA for neonatal mice. This study further confirms that the critical period of SYNGAP1 protein function is in the first two weeks of hippocampal development. In 2013, a similar study [136] by the same group demonstrated an altered critical period of development in mPFC and thalamocortical connections in *Syngap1*$^{+/-}$ during development, further validating early excitatory synaptic maturation in mPFC and, thereby, restricting the duration of the critical period of plasticity window [136]. This study further suggests that, owing to a mutation in *Syngap1*$^{+/-}$, an altered critical period of plasticity and early maturation of dendritic spine structures might prevent remapping of connections, particularly to any experience, during development. In a study by Aceti *et al.* [19], tracking of dendritic arborization in the somatosensory cortex in *Syngap1*$^{+/-}$ at PND21 unveiled several adult-like features of neurons, such as higher order branching, arborization and adult-like dendritic length, suggesting early maturation of neurons [19]. A filopodia to spine structure transition is often associated with functional remapping of sensory circuits in response to experiences [142]. Whisker deprivation indicated a 2.5-fold increase in filopodia density at PND21 in WT which was absent in *Syngap1*$^{+/-}$; this further suggests a limited capacity to organize cortical circuits or remapping cortical circuits in *Syngap1*$^{+/-}$. To further study the developmental regulation of SYNGAP1 and early maturation of neuronal spines, Aceti and group did genetic rescue by inducible Cre-allele post-critical period. Different behavioural phenotypes such as risk-taking, novelty-induced hyperlocomotion and long-term memory were rescued in neonatal mice (tamoxifen injected in PND1), whereas the young adult (PND21) counterpart showed partial rescue of brain dysfunction in *Syngap1*$^{+/-}$ [19]. Overall, these studies suggest that there is indeed a heightened period of development and plasticity of SYNGAP1 protein in different brain regions within the critical period window, which is irreversible in *Syngap1*.

## 5.2. FMR1

Fragile X syndrome is the most common monogenic form of ID [143,144]. It is an X-linked disorder [145] and, therefore, generally affects males more prevalently (1 in 4000) than females (1 in 8000) [146,147]. It is caused by the loss of function of the fragile X mental retardation protein (FMRP) [148]. The absence of FMRP is most often the result of transcriptional silencing of the locus containing the *FMR1* gene [149]. Typically, the *FMR1* gene consists of a polymorphic repeat region present in the 5′-UTR (untranslated) non-coding region of the *FMR1* gene [148,150]. It consists of a stretch of trinucleotide CGG repeats that varies between six and 54 in normal alleles [150]. Such trinucleotide repeats were found to be susceptible to expansion and contraction events during the process of DNA replication [151,152]. When this region consists of more than 200 trinucleotide repeats, it leads to epigenetic silencing owing to hypermethylation of the repeat region, and the neighbouring CpG islands present in the promoter region leading to heterochromatin formation [149,153–156]. The formation of condensed heterochromatin in this region creates a microscopically visible constriction in the corresponding site on the X-chromosome, from which the disease derives its name [148,157]. Alleles containing 54–200 CGG repeats do not lead to silencing but are more likely to expand further and, thus, are termed pre-mutations [158]. Although pre-mutations do not cause fragile X syndrome, they were found to be responsible for causing fragile-X-related disorders such as fragile-X associated tremors/ataxia syndrome (FXTAS) [159,160] and fragile X-related primary ovarian insufficiency (FXPOI) [161,162] (box 1).

Other than having an ID, patients with fragile X syndrome exhibit a range of morphological abnormalities such as macroorchidism (enlarged testis), macrocephaly, elongated face, prominent jaws and forehead, and a highly arched palate and contain loose connective tissue leading to highly extendable joints, flat feet and soft skin [163–165]. To further understand the pathophysiology and aetiology of this disorder, the most commonly used mouse model is the *Fmr1*-KO mouse, which was first developed in 1994 [166].

## 5.2.1. Behavioural changes associated with *Fmr1* mutation

Patients with fragile X syndrome display a variety of neuropsychiatric symptoms that mainly include cognitive deficits, delayed language development, hyperactivity, social anxiety, impulsivity and a subset of autistic behaviours such as stereotypic and repetitive behaviour, shyness, poor eye contact and hypersensitivity to sound. Other than ID/ASD, fragile X syndrome frequently co-occurs with additional neuropsychiatric disorders such as epilepsy, sleep disorders and ADHD [163,164,167–171].

Numerous studies have been performed to study the emergence and nature of behavioural characteristics in the *Fmr1*-KO mouse model. Various types of fear-conditioning studies that test memory (hippocampus, amygdala and mPFC dependent) have revealed mild impairments in *Fmr1*-KO mice [172–176]. Additionally, in the Morris water maze test, which requires intact long-term hippocampus-dependent memory, *Fmr1*-KO showed learning deficits during the phases of acquisition and reversal memory [166,177]. *Fmr1*-KO mice failed to discriminate during the novel object recognition task, a well-established short-term memory task [178,179]. Additionally, *Fmr1*-KO mice showed exaggerated inhibitory (passive) avoidance extinction, demonstrating impaired emotional memory processing [173,180–182]. Thus, the *Fmr1*-KO mice were found to

royalsocietypublishing.org/journal/rsob Open Biol. 9: 180265

**Box 1.** Mechanism of trinucleotide expansion.

Although the mechanism is unknown, it is thought to occur as a result of the formation of secondary hairpin loop structures by these repeats in the daughter strand. During replication, DNA polymerase synthesizes a new complementary strand of DNA along the original parent strand, which then bind to each other. When a long stretch of DNA has newly synthesized trinucleotide repeat-sequence contacts and attaches to itself due to self-complementarity, it forms a secondary hairpin loop structure, leaving the just-copied parent strand unbound. As a result of this gap, the DNA polymerase slips back and re-replicates the same sequence. In this manner, the daughter strand now contains an extra set of trinucleotide repeats, i.e. expansion takes place [151].

display various forms of memory impairments and cognitive deficits such as increased locomotor activity, anxiety, poor communication based on the ultrasound test, repetitive behaviour, defective sensory motor gating and seizures and to prefer social isolation [166,173,183–192]. Fmr1-KO mice further showed increased stereotypic and repetitive behaviours as observed from self-grooming [186,193] and marble-burying tendencies [194]. Based on these studies, it is evident that the biochemical, molecular, physiological and behavioural findings in mice are similar to phenotypes observed in human patients. Therefore, many of the Fmr1 mouse models are useful tools to study the effect of monogenic mutations in pathophysiology and synaptic deficits as well as behaviour, and can be a potential preclinical model to find novel therapeutic targets [195,196]. However, caution should be shown to determine a suitable mouse model as it should fulfil the criteria discussed in the animal model section earlier.

### 5.2.2. Synaptic function and morphology

During development, spine morphology gradually changes from thin, motile immature, elongated spines to larger, more stable, mature, stubby, mushroom-shaped spines [197–199]. Golgi staining of post-mortem brain tissue of patients with fragile X syndrome revealed dysgenesis in the spine morphology. Neurons in these tissues showed increased spine density, most of which were thin and elongated, indicating an immature spinal phenotype [200]. A similar observation was made in neurons from multiple cortical areas [201]. Fmr1-KO mice also exhibit a similar spine morphology such as the increased density of thin, long, immature spines in many cortical regions and the hippocampus [202,203]. This change in the spines may result from impaired synaptic turnover and maturation of spines during development, leading to retention of an immature spinal phenotype during the adult stages [204]. Studies have indeed found that increased spine turnover fails to decrease after the first two weeks of post-natal development and persists until four weeks of age, and even until adulthood [205,206]. The number of mature dendritic spines (those opposed to pre-synapse) is also reduced in Frmr1-KO mice [207]. This general lack of spine maturation is corroborated using electrophysiological evidence [208,209].

The morphology of spines correlates with the functional properties of synapses such as plasticity. Studies of basal synaptic transmission in Fmr1-KO mice did not show any difference in mEPSC amplitude [210], input–output (IO) curve or paired-pulse ratios (PPRs). However, a difference was observed in evoked spontaneous events, indicating that knockdown of FMRP does not change intrinsic synaptic electrophysiological properties, but reduces or delays the number of functional synaptic connections [210]. Fmr1-KO mice also showed a reduced AMPA/NMDA receptor ratio during the developmental stages [209,211]. Concerning plasticity, Fmr1-KO mice exhibited enhanced group I mGluR-dependent LTD [212]. However, no impairments were observed in NMDAR-mediated LTD [213].

On the other hand, NMDAR-mediated LTP was impaired in many cortical areas of Fmr1-KO mice. In the hippocampus, NMDAR-mediated LTP induced by theta-burst stimulation was also reduced in Fmr1-KO mice [214–218]. Both mGluR-mediated LTD and sustained theta-burst-mediated LTP require local synaptic protein synthesis, showing that FMRP may play a role in the regulation of synaptic protein synthesis. Large-scale forms of plasticity such as homeostatic plasticity have also been reported to be impaired in Fmr1-KO mice [219,220]. Other than glutamatergic synaptic impairments, Fmr1-KO mice further show reduced dopamine- [221] and GABAR-mediated signalling, and the number of GABARs were altered in Fmr1-KO mice [222]. These alterations in GABAR cause aberrant GABAR-mediated signalling, contributing to an altered E/I balance [223–226]. These studies suggest that the impaired group I mGluR-mediated protein synthesis and altered E/I balance lead to the decreased seizure threshold and learning and memory deficits observed in patients.

Impaired synaptic morphology and function may lead to abnormal neuronal circuit phenotype and subsequent behaviours. The E/I balance is altered in Fmr1-KO mice [226]. In general, reduced inhibition and increased excitation levels were observed in these mice, corroborating the increased susceptibility to seizures and epileptogenesis [90,227–229]. Thus, these alterations in the spine morphology and synaptic function could lead to ID and ASDs in human patients.

### 5.2.3. Biochemical pathways

The FMR1 gene present at locus Xq27.3 in humans codes for the FMRP protein [148]. FMRP was found to be majorly expressed in the brain and testes [230]. It consists of three RNA binding domains—two of which are K homology domains (KH1 and KH2), the third being the RGG (arginine–glycine–glycine) domain [231,232]. Thus, FMRP binds to RNA and regulates many of its dynamics. FMRP binds to almost approximately 4% of the neuronal RNA population and regulates the expression of their proteins, thereby affecting many neuronal and synaptic properties [231,232]. Once bound to mRNA, FMRP negatively regulates its translation [233]. This is hypothesized to take place by one or a combination of three mechanisms: (i) FMRP binds to secondary G-quadruplex structures in mRNA, stalling ribosomes

[234–237]; (ii) FMRP recruits CYFIP-mediated inhibition of translation [238]; and (iii) FMRP recruits inhibitory miRNA containing AGO2 (Argonaute 2) complex to the mRNA, leading to RNA-induced silencing complex (RISC)-induced silencing [239–242]. FMRP additionally has nuclear import and export signals [243]. FMRP has been reported to bind mRNA in the nucleus and regulate RNA's transport to dendritic spines [244,245]. Finally, the two tandem Agenet domains were found to recognize trimethylated serine residues, and, thereby, have been reported to interact with histones and modify chromatin dynamics in the nucleus [245,246]. It is evident from these studies that FMRP binds to different RNA domains, thereby regulating the strength of synapses in response to a stimulus.

An important physiological function of FMRP in neurons is the activity-dependent group I mGluR-mediated repression of local protein synthesis in dendritic spines [247]. FMRP, in its phosphorylated form, represses translation [248,249]. Neurotransmitter-mediated activation of group I mGluRs leads to the activation of a phosphatase PP2A which dephosphorylates FMRP [250]. This change in PP2A leads to removal of FMRP from the mRNA, enabling local protein synthesis of various plasticity-related proteins, including those involved in AMPAR endocytosis [249,251,252]. On a slower time scale, mGluR activates mTOR, which, in turn, inactivates PP2A and activates S6 kinase, leading to re-phosphorylation of FMRP and repression of translation [253,254]. Thus, knock-out of *Fmr1* leads to elevated levels of protein synthesis, causing dendritic spine dysmorphogenesis and impaired mGluR-mediated LTD [161,255]. In conclusion, such changes at the biochemical and molecular level impacts synaptic function and behaviour.

### 5.2.4. The critical period of plasticity

The critical period of plasticity is often found to be disrupted in many of the NDDs. One of the first studies in the somatosensory cortex by Harlow *et al.* [256] has shown the impairment of the critical period of plasticity in the *Fmr1*-KO mouse model. Behaviourally, *Fmr1*-KO mice showed altered sensory processing, as discussed above. Previous studies have shown an abundance of long thin immature dendritic spines in the somatosensory cortex [257,258]. However, these observations were correlative and unclear regarding the mechanism for various behavioural abnormalities. Using voltage clamp recordings, Harlow *et al.* [256] measured the ratio of NMDAR- and AMPAR-mediated current (NMDA/AMPA ratio) from the spiny stellate cells of the somatosensory cortex. In WT mice, the NMDA/AMPA ratio decreased progressively from PND4 to PND7, marking the closure of the critical period, whereas in $Fmr1^{-/Y}$ mice the NMDA/AMPA ratio increased between PND4 and P7 and returned to the WT level at PND10–14 [256]. In addition, loss of LTP induction was shown to be a manifestation of the closure of the critical period in the somatosensory cortex [63], which was delayed in $Fmr1^{-/Y}$ mice [256]. These data suggest a delay in the maturation of the thalamo-cortical synapses in fragile X syndrome, which could be the reason for impaired sensory processing, learning and memory. Along the same lines, altered ocular dominance plasticity in the visual cortex of $Fmr1^{-/Y}$ mice was shown by measuring visual evoked potentials (VEPs) [182,259]. Later, a maladaptive auditory response manifested by

patients with fragile X syndrome and in *Fmr1*-KO animals was shown to be a result of the impaired critical period of plasticity in the primary auditory cortex [229]. In conclusion, alteration in the critical period of synaptic plasticity is a significant contributor to the behavioural and synaptic pathophysiology, which may prevent neuronal remapping in the fragile X syndrome.

### 5.3. MeCP2

X-linked heterozygous mutations in *methyl CpG binding protein 2* (*MeCP2*) has been shown to cause Rett syndrome (RTT) in humans [260–263]. *MeCP2* has one of the longest known 3′-UTRs of the human genome and contains four exons [264]. It is believed that the diverse and complex function of *MeCP2* partly lies in its 3′-UTR. Studies have shown that different polyadenylation signals also bring changes in the expression pattern of *MeCP2* [265].

### 5.3.1. Behavioural alterations

Studies have shown that approximately 1 in 10 000 females is affected by RTT [266–268]. Studies on human patients between 7 and 18 months of age showed phenotypes of deteriorated higher cognitive and social functions, stagnancy of brain development, severe dementia, autism, ataxia and repetitive hand movements [266–268]. Constitutive *MeCP2* hemizygous KO and conditional KO mouse models showed motor deficits such as improper gait, hindlimb clasping, irregular breathing [98] and microcephaly [98,269], and did not survive for more than 12 weeks after birth. Apart from the motor and social impairment, a mouse model of *Mecp2* also manifested impaired hippocampus-dependent spatial memory, and contextual fear memory [270]. Post-mitotic neuron-specific KO and overexpression of *Mecp2* displayed a similar phenotype to patients with RTT. These animals showed deficits in motor behaviour, an increase in anxiety-related behaviour, impaired social interaction and alteration in learning and memory [269,271]. In addition, a C-terminal deletion mouse model of *MeCP2* also displayed a similar kind of phenotype, with impaired motor learning, social deficits and epileptic seizures [272]. All these results describe successful representation of RTT in mouse models, opening the possibility for effective pre-clinical studies.

### 5.3.2. Synaptic function and morphology

Post-mortem studies from patients with RTT revealed reduced axonal and dendritic processes and decreased dendritic spine density in CA1 pyramidal neurons [273,274]. A similar phenotype, i.e. reduced neuronal soma size and decreased dendritic arborization, was observed in the cortical pyramidal neurons of layer II/III of the *MeCP2* null mouse [275]. Thus, reduced complexity and size of the neurons could be one of the underlying reasons for the impaired behavioural phenotypes in the animal models similar to the patients. Considering the function of MeCP2, it can be speculated that dysregulation in the expression of many genes such as *Bdnf*, which is important for neuronal growth, may lead to altered neuronal morphology in *Mecp2* mutant mouse models. However, a mouse model overexpressing *MeCP2* had shown similar spine morphology to the *MeCP2* null mouse and patients with RTT. This study suggests that the

excess level of the protein affects the synaptic functions, which, in turn, may lead to learning and memory deficits [276].

Patients with RTT show cognitive deficits, which could be due to impairment in the synaptic functions [266–268]. Synaptic functions were altered in the *MeCP2* mutant mouse models, consistent with the structural and behavioural alterations [277–279], thereby causing E/I imbalance. In addition, modification in basal synaptic transmission has resulted in impaired hippocampal LTP, not only in KO but also in knockin studies [270,279,280]. It is clear from these studies that any fluctuation in the level of MeCP2 can alter not only behaviour but also synaptic function.

### 5.3.3. Biochemical pathways

MeCP2 regulates transcription through DNA methylation and histone acetylation. As the name suggests, it binds to the methylated cytosine residues on DNA through its methyl CpG binding domain (MBD) [281–283]. Another domain, the transcriptional repression domain (TRD), inter-acts with HDACs and mSin3a to regulate transcription of the downstream genes [281]. These proteins act as a corepressor of transcription. Therefore, the interactions are necessary to bring about transcription regulation through MeCP2. MeCP2 is shown to regulate many genes such as *Bdnf*, *Dlx5*, *Dlx6*, *Reln* and *Ube3A*, which are crucial for neuronal maturation and development, and the protein product is enriched in synapses [284–287].

MeCP2 is a master regulator of the transcription that regulates the neuronal maturation process by controlling the expression of the genes mentioned above. Besides its role in transcription regulation, MeCP2 has been shown to localize in heterochromatin by associating with chromatin remodelling complexes such as SWI/SNF, ATRX and histone methyl-transferase [282,288–291]. Association of MeCP2 with these different complexes allows the possibility of a global role in the regulation of gene expression. For a detailed review of the biochemical functions of MEPC2, please refer to Singh *et al.* [292] and Guy *et al.* [293].

Therefore, MeCP2, being a transcription regulator, regulates the expression of a wide array of synaptic plasticity-related genes. Hence, it modulates the neuronal/synaptic function which is crucial for healthy brain functions. However, a perturbation in MeCP2-mediated regulation due to a pathogenic mutation alters synaptic function and behaviour, resulting in RTT-related pathophysiology.

### 5.3.4. The critical period of plasticity

MeCP2 is required at different stages of brain development from increasing dendritic complexity to synaptogenesis and astrocyte maturation. Disrupting MECP2 at any of these stages has been linked to RTT-like phenotypes [294]. The maturation of the cortical GABA inhibitory circuitry, particularly parvalbumin+ (PV+) fast-spiking interneurons, is a key regulator for the initiation and termination of the critical period. MECP2 KOs exhibited accelerated functional maturation of PV interneurons, which correlated with a precocious onset and closure of the critical period and deficient binocular visual function in mature animals [99]. The findings to validate the role of MeCP2 in the critical period came from another study which observed that specific *Mecp2* deletion in GABAergic PV cells abolished the visual experience-dependent plasticity during the critical period in post-natal development of the visual cortex, while conditional *Mecp2* deletion in somatostatin-expressing GABAergic cells or glutamatergic pyramidal cells had no such effect [295]. Their study demonstrated that, during the critical period, selective deletion of the RTT-related gene *Mecp2* in GABAergic PV neurons could result in defective inhibitory PV neuronal circuits in the developing visual cortex, which leads to the absence of experience-dependent critical period plasticity [295].

## 5.4. SHANK

SH3 and multiple ankyrin repeat domain proteins (SHANKs), also known as ProSAPs, are a family of post-synaptic scaffolding proteins present in the excitatory glutamatergic synapses [296]. Being a primary scaffolding protein, SHANKs organize other proteins in the synapse. Thus, they are essential for normal neuronal development and function [297]. Mutations in different *Shank* genes (*SHANK1*, *SHANK2* and *SHANK3)*, especially *SHANK2* and *SHANK3*, are associated with ASDs [298], and co-occurrences with ID [299]. A recent meta-analysis study has proposed that around 1% of all patients with ID/ASDs had a mutation in *SHANK* genes [300]. Thus, it is vital to understand the pathophysiology of such mutations to develop the correct treatment.

### 5.4.1. Behavioural alterations

A study on the *Shank1* null mouse model showed reduced social interaction (social sniffing), reduced ultrasonic vocalization, increased self-grooming and repetitive behaviour, enhanced spatial learning and impaired fear conditioning [301,302]. Although deletions in exons 6–7 and 7 of *Shank2* mouse models showed contradictory deficits in the synaptic functions, comparable behavioural phenotypes were seen. Both the mouse models displayed increased locomotor activity, increased anxiety-like behaviour and impaired social behaviour [303,304]. However, exon 7 deletion of *Shank2* resulted in increased self-grooming but normal working memory [304].

A mouse model of *Shank3* with exon 4–9 deletion showed impaired social behaviour, reduced ultrasonic vocalization, increased self-grooming and impaired novel object recognition [107,305]. Other models of *Shank3* mutation displayed similar behavioural deficits, which are summarized in table 1. Thus, it is evident that the mouse models for different *Shank* mutations can represent many of the behavioural deficits observed in humans.

### 5.4.2. Synaptic function and morphology

Mutation in the PDZ domain of *Shank3* showed reduced dendritic spine density during maturation, whereas mutation in the *Ank* and *Sh3* domains led to a reduced spine head volume [306]. Therefore, SHANK3 plays an essential role in dendritic spine maturation and morphology. An *in vitro* study using cultured rat neurons showed impairment in mGluR5-dependent plasticity and signalling, demonstrating the importance of SHANK3 downstream to mGluR5-mediated signalling [307]. A mouse model of *Shank1* exhibited reduced basal synaptic transmission, whereas LTP and

LTD were not altered [302]. On the contrary, opposing effects were observed when exon 6–7 and exon 7 were deleted from *Shank2*, suggesting that there may be a possibility for an isoform/transcript-specific function of this gene, but it is unclear whether splice variation has any implications for human patients. A study by Schmeisser *et al.* [304] showed reduced spine density and synaptic transmission in the CA1 region of the hippocampus when exon 7 was deleted. In addition, a decreased I/O ratio and mEPSC frequency were seen in the exon 7 deletions, suggesting an impairment in pre-synaptic neurotransmitter release upon exon 7 deletion. I/O is the measure of basal synaptic transmission, indicating a corresponding output to every input provided to a neuron. In contrast to the increased NMDA/AMPA ratio in exon 7 deletion, decreased NMDA/AMPA was observed in the exon 6–7 deletion model of *Shank2* mutation [303,304]. Again, these observations show that the different exons in the gene might have opposite effects on the synaptic functions.

A *Shank3* mouse model with exon 4–9 deletion had severe defects in synaptic function. For example, activity-mediated spine remodelling was impaired in the hippocampal CA1 region, leading to learning and memory dysfunctions. AMPAR functions were affected in these mouse models, which was manifested by reduced AMPAR-mediated basal transmission. However, the study showed that the decreased mEPSC amplitude was accompanied by a subsequent increase in mEPSC frequency. The increase in mEPSC frequency suggests an impairment in the level of pre-synaptic function or an increase in the number of functional synapses. Owing to impaired AMPAR-mediated synaptic function, LTP was reduced significantly. These results can be corroborated with impaired learning and memory observed in patients. However, no change in NMDAR- and mGluR-mediated LTD was observed in these mouse models [107,305]. These data reiterate the fact that SHANK3 plays an essential role in synaptic signalling and function. Also, the data indicate the role of SHANK3 not only in mGluR-mediated signalling but also in AMPAR-mediated synaptic transmission. Several other mouse models of *Shank3* show deficits in different synaptic functions and are summarized in table 1.

### 5.4.3. Biochemical pathways and signalling

SHANK proteins are postsynaptic scaffolding proteins associated with the PSD complex in excitatory glutamatergic synapses [308–310]. Structurally, the SHANK family of proteins consists of five distinct motifs/domains: ankyrin repeat domain (ANK), Src homology 3 (SH3) domain, PSD-95/disc-large/ZO-1 (PDZ) domain, proline-rich (Pro) and sterile alpha motif (SAM) domain [311]. Studies that have taken place over decades have identified at least 30 proteins interacting with SHANK proteins, including different receptors, ion channels, cytoskeletal proteins and signalling molecules [297,307,308,312–314]. A study on *Shank3* KO showed an alteration in the mGluR5-HOMER scaffolding that in turn affected the neuronal connections in the brain [315]. However, knockdown of *Shank3* led to decreased mGluR5-mediated phosphorylation of ERK1/2 and CREB [307]. These independent observations link the importance of SHANK3 in mGluR5-mediated signalling and, in turn, are associated with regulation of synaptic functions.

Mouse models of the *Shank1* mutation showed a decreased synapse-associated protein-90/postsynaptic density-95 associated protein (SAPAP), guanylate kinase-associated protein (GKAP) and HOMER protein level in the PSD complex [302]. Exon 6–7 deletion in the *Shank2* model showed reduced phosphorylation of calcium/calmodulin-dependent kinase II (CaMKII), extracellular signal-regulated kinase (ERK) and the AMPAR subunit, GluA1, but increased expression of the NMDAR subunit, GluN1 [303]. Deletion of exon 7 from *Shank2* increased NMDAR subunit, GluN2B, expression in the hippocampus, whereas, in the striatum, GluN1, GluN2A, SHANK3 and GluA1 levels were increased [304]. These data show that there could be a different function for SHANKs in different brain regions. Also, by modulating the expression of different NMDAR subunits, synaptic properties involved in learning and memory were modulated by different SHANKs. Studies from a mouse model of *Shank3* with exon 4–9 deletion showed a robust reduction in the level of GluA1 subunit [107,305]. *Shank3* knockdown in cultured hippocampal neurons showed reduced expression of mGluR5 but not NMDAR or ERK [307], whereas analysis from the striatal PSD fractions of the *Shank3B*$^{-/-}$ mouse model showed a decreased level of NR2A, NR2B, GluA2 and HOMER [108]. However, other mouse models of *Shank3* also show related biochemical alteration (table 1). Based on the studies above, it is evident that the SHANK family of proteins have diverse roles in cellular signalling at different brain regions. In conclusion, the SHANK family of proteins differentially regulate synaptic function by modulating different receptor subunit expression, and, thus, regulate neuronal function, learning and memory, and behaviour.

### 5.4.4. The critical period of plasticity

An *in vivo* study has shown that loss of *Shank3* led to impairment in the ability of visual cortical circuit recovery following sensory input deprivation [316]. Also, the homeostatic plasticity of neuronal circuits was disrupted in the *Shank3* KO model, which hints towards perturbation of the critical period in *Shank3* mutation [316]. However, using a conditional knockin mouse model, Mei *et al.* [317] showed re-expression of SHANK3 in adulthood restored spine density and synaptic functions in the striatum. Repetitive grooming and impaired social interaction were improved, whereas no improvement was seen in anxiety-like behaviour and motor behaviour [317]. From these studies, we can speculate that there was no major effect of *Shank3* perturbation on the critical period of development. However, another possibility could be delayed closure of the critical period, which may persist until adulthood in the *Shank3* KO, but this is unclear. This may explain the partial rescue of behavioural phenotypes observed after restoring *Shank3* expression in adulthood.

## 5.5. Neuroligins

For almost two decades, it has been known that, to hold a particular synapse together for communications between neurons to occur, there are synaptic adhesion molecules such as NEUROLIGINS present in the post-synapse [318,319], and NEUREXINS, present in the pre-synapses, are known to hold a particular synapse together [320–322]. Mutations in *NEUREXIN* and *NEUROLIGIN* genes increase

royalsocietypublishing.org/journal/rsob   Open Biol. **9**: 180265

royalsocietypublishing.org/journal/rsob Open Biol. 9: 180265

the likeliness of affecting synapse formation, as well as function [318,319,323,324]. *NRXN1, NRXN2* and *NRXN3*, in mammals encode for NEUREXINS, α-NEUREXIN and β-NEUREXIN, depending on their promoters [320,321]. NEUROLIGINS generally interact with β-NEUREXIN isoforms [318]. NEUROLIGINS, on the other hand, are encoded by five genes, *NLGN1, NLGN2, NLGN3, NLGN4* and *NLGN4Y*, in humans [318,319,325,326]. Based on immunostaining and biochemical analysis, subcellular localization of NLGN1 was found to be present at the excitatory synapses, and the expression level is low at birth but increases during post-natal days 1–8 and remains relatively high in later stages of development in mouse [323]. NLGN2 and NLGN4 were shown to be expressed in inhibitory synapses [327,328], whereas NLGN3 was expressed in both excitatory as well as inhibitory synapses [329]. With the help of different binding partners, such as PSD-95 [330], MAGUK [331] and GluN1 [332] at excitatory synapses and GEPHYRIN [242,327] and COLLYBISTIN [333,334] at inhibitory synapses, all NEUROLIGINS are required to maintain synapse number/density and to regulate maturation and differentiation of synapses.

NEUROLIGINS were recognized to induce the formation of functional synapses in early 2000. These studies, using non-neuronal cells expressing NEUROLIGIN, revealed that NLGN1 and NLGN2 alone could trigger the formation of pre-synaptic structures such as clustering of synaptic vesicles in the axon terminals of central nervous system (CNS) neurons [335]. The role of NEUROLIGINS was further validated by dissecting functional characteristics, such as NMDAR-mediated excitatory postsynaptic currents and NMDAR-dependent LTP, which are a cellular correlate of learning and memory [336,337]. This study suggests a vital role of NEUROLIGINS in learning and memory. Moreover, Varoqueaux *et al.* [338] have shown that the *Neuroligin* mouse model dies after birth due to a reduced neuronal network activity and reduced glutamatergic and GABAergic synapse formation/function resulting in respiratory failure [338], suggesting the importance of NEUROLIGINS not only in synapse formation and maturation but also in neuronal function. *In vitro* electrophysiological studies have shown altered mIPSC amplitude as well as frequency, suggesting a reduced number of functional GABA receptors at the post-synapse resulting in altered E/I. Furthermore, decreased excitatory and inhibitory synapses were observed in the downregulation of NLGN1 and NLGN2 [339]. The introduction of exogenous NLGN increased both mEPSCs and mIPSCs indicates an increase in excitatory and inhibitory synaptic function (increased number of functional AMPAR and GABAR in the post-synapse, respectively) [340], implying the critical role of NEUROLIGINS in maintaining E/I balance.

Clinically, *NLGN1* genetic variants are associated with disorders, such as ASDs [341,342], Alzheimer's disease (AD) [343] and post-traumatic stress disorder (PTSD) [344]. Genome sequence studies in humans have shown dysfunction in *NLGN2* to be associated with ASD and schizophrenia. Studies were done by assessing the developmental history of a patient with a rare missense mutation, R215H, which revealed that the patient had psychotic symptoms such as self-laughing and talking, auditory hallucinations and delusions. However, the patient's sibling was a carrier of this mutation, suggesting that the R215H mutation is inheritable and had incomplete penetrance [345,346].

### 5.5.1. Behavioural alterations

Impaired spatial working memory by using the Morris water maze in either loss of *Nlgn1* or overexpression of *Nlgn1* was observed, which further implies the constitutive requirement of *Nlgn1* for learning and memory [347,348]. Conditional KO of *Nlgn1* in the CA1 region of the hippocampus in new-born (P0) or P21 resulted in impaired NMDAR-type and L-type $Ca^{2+}$ channel-dependent LTP, further validating the loss of spatial working memory [349]. *Nlgn1* KO mice created by targeted deletion of exon sequences covering the translational start site 380 bp of the 5′ coding sequence of NLGN1 and by homologous recombination in embryonic stem cells exhibited increased repetitive behaviour such as grooming, impaired social interaction and altered pain sensation [347].

Overexpression of *Nlgn2* in transgenic mice had displayed diverse behavioural deficits such as reduced lifespan, limb clasping, offspring viability, repetitive behaviour, anxiety and impaired social interactions [350]. Based on the inhibitory avoidance (IA) behaviour paradigm, widely used for studying fear memories, Ye *et al.* [351] had found increased expression of NLGN1 and NLGN2 in quantitative immunoblot analyses after training of rats for IA. This altered expression of NLGN1 and NLGN2 suggests a role for both *Nlgn1* and *Nlgn2* in memory consolidation [351]. Further studies were performed to understand the role of *Nlgn2* in memory formation and behaviour. Overexpression of *Nlgn2* in the hippocampus had shown an increase in adult neurogenesis but decreased performance in the water maze task, suggesting an impaired working memory [352]. *Nlgn2* KO mice showed reduced anxiety, increased impulsivity in the elevated plus maze and reduced fear conditioning with an increased ratio of evoked E/I synaptic currents [353,354]. These reports proved the importance of *Nlgn2* in diverse behavioural functions by regulating inhibitory synapse function and plasticity in the mPFC, which is essential for anxiety and fear memory. In addition, *Nlgn2* KO mice exhibited an irregular breathing pattern, suggesting its role in regulating lung and heart functions [338].

As discussed in the earlier section regarding the significant role of *Nlgn4* KO in synaptic function, other studies have shown perturbations in general behavioural patterns, such as visible platform training in the Morris water maze (for vision), buried food finding (for olfaction), sucrose preference (taste), startle response (hearing), prepulse inhibition (sensorimotor gating), rotarod (locomotor activity and balance), hole board (exploratory behaviour), object preference, open field, hidden platform training in the Morris water maze, cued and contextual fear conditioning, and reversal training in the Morris water maze but lacked seizure propensity. These studies demonstrate a selective deficit in social interaction in *Nlgn4* KO as seen in patients with ASDs [109,355,356].

### 5.5.2. Synaptic function and morphology

A transgenic mouse model of *Nlgn1* overexpression led to an increase in excitatory dendritic spine and synapse number, E/I ratio and synaptic transmission in the hippocampus. Additionally, overexpression or downregulation of *Nlgn1* has been shown to have impaired long-term potentiation (LTP), suggesting importance for *Nlgn1* in learning and memory [348,357]. Moreover, studies from *Caenorhabditis elegans* lacking *nrxn-1* or *nlgn-1* have been shown to mediate retrograde synaptic signalling that inhibits neurotransmitter

release at neuromuscular junctions, which might affect the activity of neurons in response to stimuli [358]. *Nlgn1* KO mice made by targeted deletion (exon sequences covering the translational start site 380 bp of the 5′ coding sequence of *Nlgn1*) by homologous recombination in embryonic stem cells showed no change in PPRs, and basal synaptic transmission but altered LTP and AMPA/NMDA [347]. These studies from different mouse models show that *Nlgn1* is necessary for AMPAR dynamics and, thus, the alteration would impair learning and memory.

E/I imbalance is one of the significant characteristic features of ID and ASDs that causes various physiological and behavioural deficits. In fact, in *Nlgn2* transgenic mice, the E/I ratio was found to be decreased in the PFC along with the increased frequency of miniature inhibitory synaptic currents, which suggests an inclination towards potentiation of inhibitory synapses and, thereby, shifting towards altered inhibition [350]. *Nlgn2* knockdown was associated with abolished GABAergic function from excitation to inhibition switch in cortical neurons based on $Ca^{2+}$ imaging studies that show a gradual decrease of GABAR-evoked $Ca^{2+}$ response in developing neurons along with the decreased frequency of mIPSC and mEPSC, suggesting a reduction in excitatory and inhibitory receptors. However, overexpression of KCC2, the potassium-chloride co-transporter, partially rescued synaptic currents, suggesting a role of NLGN2 in excitatory as well as inhibitory synaptic function [359].

Along with these earlier studies, another study has reported that *Nlgn2* KO mice have developmental delays such as delayed eye-opening period, vocalization in pups and reduced body length in these *Nlgn2* KO mice [360]. Furthermore, these mice displayed diminished inhibitory synaptic transmission with no change in synapse number in the ventrolateral medulla [333]. These studies suggest that *Nlgn2* is an essential constituent of inhibitory synapses and necessary for the formation, maintenance and function of inhibitory neurons.

To further study the role of *Nlgn3* mutations in synaptic function and neuronal development *in vivo*, lentiviral-mediated knockdown of *Nlgn3* in the CA1 region of the hippocampus at P0 and P21 was performed, and excitatory basal synaptic transmission was unaffected [361]. On the contrary, a primary neuronal culture study has shown that overexpression of *Nlgn3* increased inhibitory postsynaptic currents (IPSCs), suggesting an increase in expression of GABAergic currents [362]. Patch clamp recordings from the hippocampus, somatosensory cortex and cerebellum of *Nlgn3* KO mice have revealed increased mIPSC (increased number of functional GABA receptors) and decreased mEPSC (decreased number of functional AMPA receptors) in the hippocampus, suggesting an increase in inhibitory activity as the number of functional GABARs was higher when compared with excitatory activity. Additionally, decreased mEPSC and impaired mGluR-mediated LTD in the cerebellum was also observed, which suggests a differential function of *Nlgn3* in different regions of the brain [110,363].

An *Nlgn3*, wherein R451C was overexpressed, increase in mIPSC frequency (increase in the number of functional GABARs) was associated with altered GABA release probability, concomitant with an increase in giant depolarizing potential (GDP), a neuronal network-related activity mediated by GABAR [111]. These studies further validate the importance of *Nlgn3* in network-based activities in immature neurons. Since GABAR-mediated network activity, as

well as GABAergic synaptic transmission, was impaired in R451C, it is essential to understand whether the effects observed were a global GABAR-mediated phenomenon or any specific GABAergic cell types involved in it. Using a paired whole-cell patch clamp between one PV expressing basket cells and either spiny neurons/pyramidal neurons, IPSCs measured from PV neurons displayed impaired amplitude and frequency in the hippocampus as well as in the barrel cortex in *Nlgn3* R451C knockin mice [112,113]. In addition, these studies found that the IPSC was reduced, leading to an altered E/I ratio in D1-medium spiny neurons (D1-MSN) in R451C mutant mice [114]. These studies show that the NLGN3 mutation has a stronger effect on PV cells comparatively, and is vital for D1-MSN-mediated synaptic transmission.

Moreover, the same knockin mouse model of *Nlgn3*, R451C, showed increased dendritic complexity and branching in stratum radiatum. These mice also showed increased excitatory basal synaptic transmission, LTP, NMDA/AMPA and mEPSC in the CA1 region of the hippocampus, suggesting, unlike KO models of *Nlgn3*, R451C function majorly affects glutamatergic synapse [110]. In contrast to previous studies, another mouse model of *Nlgn3*, R704C, displayed a decreased AMPAR-mediated synaptic response, rendering the NMADR- or GABAR-mediated response unaltered. Additionally, NMDAR-mediated LTP was associated with reduced EPSC frequency and increased NMDA/AMPA ratio in cultured hippocampal neurons, suggesting that an R704 mutation affects differently the inhibitory as well as excitatory synapses [115]. Overall, different mutant mouse models of *Nlgn3* depict region- and synapse-specific function. Therefore, specificity in neuronal types and synapses may help *Nlgn3* to execute different functions efficiently in various parts of the brain.

*Nlgn4* is another essential gene that plays a significant role in synapse formation, development and function. To further understand the role of *Nlgn4* in neuronal function, Jamain and group [355] have demonstrated reduced ultrasonic vocalizations in *Nlgn4* KO males. The *Nlgn4* KO is the result of chimeric non-functional protein. It contains a small fraction of the esterase domain that cannot bind to *Nrxn* upon contact with a female in oestrous cycle, suggesting a lack of ability to attract the opposite gender despite being fertile. A magnetic resonance imaging volumetric study demonstrated a reduction in total brain volume, particularly in the cerebellum and brainstem [355]. *Nlgn4* KO showed a reduced decay in glycinergic mIPSC, impaired inhibition, altered firing and decreased β-wave amplitude in retinal cells, demonstrating that *Nlgn4* localizes to glycinergic post-synapses and plays an essential role in encoding stimuli in the retinal network [328]. The *Nlgn4* R87 W mutation abolished NLGN4-induced synapse formation and function, particularly in modulating synaptic strength [364]. An important hallmark of several NDDs, such as ID/ASDs, is impaired synapse formation and function. As *NLGN4* is implicated in ASDs (reviewed extensively in [356]), a point mutation in *Nlgn4* causes ID/ASDs through the loss of function. A study identified a frameshift de novo mutation, 1186insT in *NLGN4*, in two siblings with ASD and Asperger syndrome in a Swedish family, and that linked *NLGN3* and *NLGN4* to ASDs. Another Swedish family of two siblings with ASD and Asperger syndrome were identified with a C to T transition in *NLGN3*. This mutation had led to changes in highly conserved arginine to cysteine (R451C), an integral

royalsocietypublishing.org/journal/rsob    Open Biol. 9: 180265

part of the esterase domain that is necessary for interaction with *NRXNs* [365]. Therefore, these studies have demonstrated that mutations in different *Nlgn* genes affect the basic synaptic transmission, which can have lasting implications for the pathophysiology in ID/ASDs.

### 5.5.3. Biochemical pathways

NEUREXIN and NEUROLIGIN contain an intracellular PDZ-binding domain that mediates interactions with synaptic scaffolding proteins, such as calcium/calmodulin-dependent serine protein kinase (CASK) [366], and Munc18 interacting protein; lin10/X11 [367] (for NEUREXINS) and PSD95 (for NEUROLIGINS) [330]. The synaptic function of NEUROLIGIN depends on *cis* clustering of NEUROLIGIN molecules, which requires a crucial integral esterase like ectodomain that can interact with NEUREXIN and execute synaptic functions [368]. PSD-95, a critical scaffolding protein interacting with NEUROLIGIN, recruits different synaptic protein/receptors like NMDAR, which activate the downstream signalling necessary for learning and memory. It also recruits specific adaptor proteins such as GKAP [369] that, in turn, interacts with SHANK3 [356,369]. This interaction is necessary for normal synaptic functions and is indirectly driven by different synaptic partners. Sequence homology studies have revealed that NLGN1, NLGN3 and NLGN4 were similar, unlike NLGN2, which is predominantly expressed in inhibitory synapses [318,319]. NLGN2 was shown to stabilize inhibitory synapses with the help of scaffolding protein GEPHYRIN, which interacts with an $\alpha2$ subunit of GABARs [370,371] and helps in the maintenance and function of GABAergic synapses. Overall, NLGN4 is considered as one of the primary receptors involved in learning and memory formation, along with evolutionary conserved other NLGNs.

### 5.5.4. The critical period of plasticity

During brain development, NLGN1, NLGN2, NLGN3 and NLGN4 expression increases from embryonic to post-natal days before reaching a plateau around three weeks in the mouse hippocampus [355]. This study suggests that NLGNs have a precise developmental window of expression and, thus, could be considered as one of the determinants of a critical period of development in the brain, which could have implications in ID/ASDs [356]. To further corroborate NLGNs in the critical period of development, monocular and binocular deprivations (MD, BD) in mice were performed. Increased spatial acuity by measuring visually evoked potentials was observed in young adult R451C *Nlgn3* mutant mice. Immunohistochemical analysis revealed an increased number of puncta of GAD65. This increase suggests an alteration in the number of GABAergic interneurons, thereby resulting in elevated inhibition that led to decreased E/I, and, thus, increased acuity in mutant mice. R451C also showed permanent loss of acuity on prolonged MD as WT, but the relative loss was more for the mutant mice, suggesting a longer window for plasticity. This alteration observed in MD might lead to the abnormal opening of a critical period of plasticity and impaired local circuit connections [355]. Studies have further tried to dissect the mechanism of NEUROLIGIN functions in synapse development and neuronal functions using different animal models such as KO, knockin and transgenic mice.

## 6. Metaplasticity

Metaplasticity is a term which refers to the higher order of synaptic plasticity, i.e. plasticity of the synaptic plasticity [372]. It includes processes that lead to physiological and biochemical changes, altering the neuron's ability to induce and maintain synaptic plasticity [373]. Different mechanisms have been proposed to explain metaplasticity depending on the location and the type of synapse or receptors of interest [373]. To date, there are no reports stating the direct involvement of metaplasticity in any gene mutations implicated in ID or ASDs. However, it has been proposed that synaptic defects and memory deficits associated with ID and ASDs may be due to an inability to undergo metaplasticity during various developmental stages [374]. It has also been demonstrated that hippocampal metaplasticity is required for the formation of temporal associative memories [375]. Although studies done to investigate the effect of metaplasticity in animal models are limited, owing to its role in the maintenance of LTP and LTD it may be crucial for learning and memory impairments in ID.

## 7. Glial cells in intellectual disability

Other than neuronal cells, the brain consists of non-neuronal cells called glia. The three main types of glia are—oligodendrocytes (responsible for axonal myelination), microglia (responsible for immunity-related functions) and astrocytes (responsible for maintaining homeostasis in the brain) [376].

### 7.1. Astrocytes

Astrocytes are star-shaped cells initially only known to provide neuronal support. However, research over the past few decades has increasingly revealed the importance of astrocytes, and their multifaceted role in the brain [20,377,378]. Astrocytes regulate post-natal neurogenesis [379,380] and maintain homeostasis of many factors such as ions, neurotransmitters, water and extracellular matrix [381–384]. They regulate many aspects of the blood–brain barrier [385,386] and glucose metabolism (supply and storage) in the brain [387,388]. They protect against damaging factors such as pathogens, reactive oxygen species and excitotoxicity [389–391]. Most importantly, astrocytes wrap around synapses and finely regulate all aspects of synaptic dynamics such as formation, maturation, plasticity and even elimination [378,392–395].

Proteins of many genes implicated in ASDs and ID are expressed in astrocytes. *Fmr1* is one such gene that encodes FMRP. In astrocytes, FMRP is expressed during early and mid-post-natal developmental stages, indicating its probable role in brain development and fragile X pathology [396,397]. Indeed, studies have shown that *Fmr1*-KO astrocytes were able to induce pathogenic delayed maturation and synaptic protein expression in WT neurons. Vice versa, WT astrocytes can rescue abnormal spinal and dendritic phenotype in *Fmr1*-KO neurons [398,399]. Astrocyte-specific knockdown of *Fmr1* led to increased spine density in cortical neurons and can be rescued by restoring astrocytic FMRP levels [21]. A study on MECP2 has also been shown to be expressed in astrocytes, and similar co-culture studies demonstrate astroglia contribution to disease pathology

royalsocietypublishing.org/journal/rsob    Open Biol. **9**: 180265

**Table 2.** Upcoming therapeutic approaches for the treatment of ID. This includes drugs which have shown some promise in the preclinical studies for the treatment of ID. HMG-Co-A, 3-hydroxy-3-methyl-glutaryl-coenzyme A reductase.

| s.no. | drug | known mechanism of action | structure | approval status | target in ID | references |
|---|---|---|---|---|---|---|
| 1 | lovastatin | statins (HMG-Co-A inhibitor) |  | approved for hypercholesterolaemia | FMRP and SYNGAP1-mediated RAS-ERK1/2 activation | [32,34] |
| 2 | rapamycin | mTOR inhibitor |  | approved as an immunomodulator | the modulator of SYNGAP1-mediated mTOR signalling | [33] |
| 3 | ganaxolone | the positive allosteric modulator of GABA$_A$ receptors |  | phase 3 clinical trial for CDKL5 deficiency disorder | the positive modulator of FMRP-mediated GABA$_A$ receptor expression | [30] |
| 4 | valproate | inhibitor of GABA transaminase and that of voltage-gated Na$^+$ channels |  | approved for mania and epilepsy | rescues from the SHANK-3 overexpression-mediated manic-like behaviour | [35] |
| 5 | oxytocin | peptide hormone which plays a role in milk ejection | Cys – Tyr – Ile – Gln – Asn – Cys – Pro – Leu – Gly – NH$_2$ | approved for improvement or facilitation of uterine contractions during birth | reverses the behavioural deficits in *Shank*-3-deficient mice | [30] |
| 6 | D-cycloserine | an antibiotic which is an inhibitor of the bacterial cell wall synthesis |  | approved antibiotic for the treatment of tuberculosis | reverses the repetitive and stereotyped grooming behavioural deficits in Neuroligin 1 knockout mice | [36] |

[400,401]. Other than these proteins, only NEUROLIGINS are expressed in astrocytes, where they contribute to the formation of astrocyte morphology and neuronal synapses [402]. Disease-related studies for *Neuroligins*, *Shank* and *Syngap1* remain to be carried out.

## 7.2. Microglia and oligodendrocytes

Microglia are the resident macrophages of the brain. They are of mesodermal origin and migrate into the brain during development [376,403]. Other than immune surveillance, microglia help in synaptic pruning (through complement-mediated phagocytosis or TREM2 signalling), clearance of apoptotic neural debris during development and synaptic plasticity (through secreted factors) [404,405]. On the other hand, oligodendrocytes originate from neural tissue and differentiate from oligodendrocyte precursor cells. They differentiate soon after neuronal migration and start wrapping neuronal axons with lipid-rich myelin, providing insulation and dividing the axon into multiple domains. The altered activity helps in regulation of the propagating action potentials and, in turn, neuronal circuit properties [406]. Oligodendrocytes also provide metabolic support and maintain ion homeostasis of the axon [407,408]. Not many studies have been performed on the relationship between genes implicated in ID/ASDs and microglia/oligodendrocytes. However, FMRP has been demonstrated to express in both cell types [397]. Further studies regarding how these mutations (implicated in ID and ASDs) affect microglia and oligodendrocytes, and how this, in turn, contributes to the pathophysiology of these disorders remains an open and exciting topic for investigation.

## 8. Current and upcoming therapeutic options targeting synaptic abnormalities

A therapeutic intervention targeting the underlying causes of ID and ASDs is not yet available in the clinic, but preclinical trials are going on in many laboratories. Drugs are, nevertheless, used to provide symptomatic relief from anxiety, epilepsy, depression, and cognitive and social dysfunctions in ID/ASDs. Response to already existing treatment options is known to be variable and is reported to have some side effects owing to their off-target interactions [29,30]. Coadministration of different medications with irrational prescription and use was another concern that affects people with ID/ASDs. More often than not, polypharmacy is a common practice and may lead to adverse drug reactions [31,32].

Although some of the currently followed strategies have prolonged the life of patients, their quality of life has not improved substantially [33]. Hence, there is a need to search for a new therapeutic intervention which can provide a cure, or at least alleviate the symptoms better than the existing therapeutic strategies for patients with ASDs and ID.

Different mutations, as discussed earlier, and drugs targeting these had undergone rigorous testing in preclinical trials on different mouse models. These have provided insights into the underlying mechanisms associated with the efficacy of these drugs for the treatment of ID/ASDs. Some of these approaches are summarized in table 2. However, their translational success is yet to be validated in clinical trials.

## 9. Conclusion

ID and ASDs are prevalent NDDs that have proven to be complicated and challenging in several aspects. Since they often present with a highly variable and overlapping spectrum of symptoms and syndromes, defining a distinct set of diagnostic criteria has been difficult for clinicians and scientists. However, studies in mouse models of monogenic causes of ID and ASDs have proved to be immensely helpful in the construction of the pathophysiology of these disorders, through a bottom-up approach. These studies have demonstrated that ID/ASDs with diverse causal origins have intersecting aetiologies that might be responsible for the observed shared phenotypes. Opposing cellular phenotypes observed in these disorders highlight the importance and need for balanced and timely developmental processes at all systemic levels.

Further studies have employed these aspects for the development of genetic and pharmacological therapeutic strategies (creation of mouse models with counteracting mutations to re-establish balance and the testing of drugs targeting common neurological pathways). However, only a fraction have been uncovered in the understanding of these disorders, and require further studies for improved diagnosis, treatment and prognosis. As highlighted in this review, there are several areas which remain unexplored and could play an essential role in the pathophysiology of ID/ASDs. Additionally, the role of non-neuronal cells such as astrocytes, oligodendrocytes and microglia have also not been studied in detail concerning mutations in ID/ASDs. The augmented critical period is another characteristic modality altered in many forms of ID/ASDs. The study of precise mechanisms for a better understanding of this phase of development could be useful for rescue during the later period of life. Because the diagnosis is delayed during a critical period of development, reversing neuronal connections becomes difficult, which is one of the significant questions lingering in the minds of neuroscientists. Preclinical studies in this regard can warrant some useful clues for the translational success of small molecules being tested for efficacy in ID/ASDs. However, many drugs still fail in clinical trials even after ameliorating disease pathology in the preclinical mouse models. Poor experimental design with inadequate sample size could be one of the reasons for failure at the later stages.

Another critical point is the variability in the intrinsic metabolic and biochemical pathways among different animal strains and species that lead to changes in drug pharmacokinetics and pharmacodynamics across systems. These factors influence how a potential therapeutic candidate molecule can be metabolized by the animal model, and how this is different in human beings. One viable alternative to overcome the above issues is to use patient-derived IPSCs, which have been considered as a model in the last decade or so. However, to acquire an all-round understanding of ID/ASDs, it is crucial to study *in vivo*, i.e. animal models in combination with patient-derived IPSCs. Such a combinatorial study can fulfil the existing gap in our knowledge about ID/ASDs and show the way for future therapeutic strategies.

Data accessibility. This article does not contain any additional data.
Authors' contribution. V.V., A.P., A.A.V. and B.V. contributed equally to this review. V.V. wrote the sections related to NEUROLIGINS and the critical period of development; A.P. wrote the sections related to MECP2 and SHANK mutation, A.A.V. wrote the sections related

to the FMRP, aetiology, contribution of non-neuronal cells and conclusion part of the review. B.V. wrote the sections pertaining to the introduction, SYNGAP1, animal models of ID, metaplasticity and therapeutic approaches. All authors contributed equally to the figures and tables. J.P.C. edited the manuscript. All the authors provided the conceptual idea for the review.

Competing interests. All authors declare no conflict of interest.
Funding. The work was supported by grants to J.P.C. by DST-SERB (SERB/JC/4518), DBT-JNCASR (LSRET-JNC/JC/4531) and intramural funding from the Jawaharlal Nehru Centre for Advanced Scientific Research, Bangalore, India.

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
