## [Reviewer comments · Open Biology]

Review History

RSOB-18-0265.R0 (Original submission)

Review form: Reviewer 1

Recommendation

Major revision is needed (please make suggestions in comments)

Are each of the following suitable for general readers?

a) **Title**
Yes

b) **Summary**
Yes

c) Introduction

Yes

Is the length of the paper justified?

No

Should the paper be seen by a specialist statistical reviewer?

No

Is it clear how to make all supporting data available?

Not Applicable

Is the supplementary material necessary; and if so is it adequate and clear?

Not Applicable

Do you have any ethical concerns with this paper?

No

Comments to the Author

This review article assesses current mouse models of ID and Autism by a junior investigator who has made contributions to the field. Overall the topic is of interest to a broad swath of neuroscientists interested in autism and synaptic mechanisms. However the manuscript needs significant editing to make it easily digestible and informative to both those in and outside of the discipline. Minor concerns are about some of the grammatical issues and awkward sentence structure which would benefit from some editorial input. But more importantly the manuscript is too diffuse without a central unifying concept. It is not clear whether the authors wish to find some convergence across the models they discuss and in fact it remains unclear why these are the focus of the article. The authors need to articulate a critical central theme and discuss the relevant models to highlight these.

Moreover the current version reads like a laundry list of every study done in these mouse models and does not critically assess or summarize key findings. This makes the manuscript a little tedious to follow and does not put forward a more thoughtful discussion of the current state of the field. For instance many behavioral tests carried out in the Fmr1 ko mice are listed and then it is stated that other studies cannot replicate these. The reader is left wondering what is the important finding in these mice. Some further assessment and conclusions are warranted in each case. This would also allow the authors to reduce the length of the manuscript which is currently overly lengthy.

The authors should also carefully consider what they want to discuss in the introduction. There is a lengthy discussion of critical periods and then this is not mentioned later in the context of mice. The thematic goals need to be more strongly tied together so that the reader can follow the line of thought and what are the important findings and outstanding gaps when trying to relate mouse studies to the human phenotype. For instance a discussion of critical periods and learning and memory don't really make much sense here.

Overall a major revision is required to improve this manuscript for publication.

Review form: Reviewer 2**Recommendation**

Major revision is needed (please make suggestions in comments)

Are each of the following suitable for general readers?

- a) **Title**
Yes
- b) **Summary**
Yes
- c) **Introduction**
Yes

Is the length of the paper justified?

Yes

Should the paper be seen by a specialist statistical reviewer?

No

Is it clear how to make all supporting data available?

Not Applicable

Is the supplementary material necessary; and if so is it adequate and clear?

Not Applicable

Do you have any ethical concerns with this paper?

No

Comments to the Author

The review "Understanding Intellectual Disability and Autism Spectrum Disorder from common mouse models: synapses to behaviour" by Verma et. al. highlighted recent advancement, upcoming challenges and pressing questions in the field. The review provided thorough and adequate information related to current understanding of ID and ASD. I strongly believe that the review will attract necessary attention in the field. However, the review needs a major revision related to presentation of existing knowledge, English language and figures. The manuscript is a good fit for publication in Open Biology with revision. My concerns are as follows:

- 1) The review needs to be threaded together for better appreciation of current knowledge in the field. Authors need to connect physiological relevance (electrophysiology data, spine defects and behavioural deficits) of the data with mechanistic details. In current version, these are presented in isolation.
- 2) Metaplasticity and contribution of glial cells in ID and ASD should be presented along with the section describing ID or ASD.
- 3) Conclusion section should highlight specific challenges, technical limitations and pressing questions in each of these fields. It should provide information about future direction of research in each of these disorders.
- 4) I noticed authors often used colloquial terminology all throughout the manuscript. Few examples – line # 46, 47, 55 in Page 2, line # 114 in Page 4, line #121 in Page 5, line 248 in page # 9, Page # 5 line 128 ..
- 5) Authors should specify specific brain area / cell type where differential expression is reported. Please see line # 55 in Page # 2 "differences in the tissue level localisation.." which tissue? Line # 58, page # 3 "SYNGAP1 is localized to the brain areas" which brain area?

- 6) Authors should mention how astrocyte maturation influence excitatory synapse development and why this process is linked to ID.
- 7) Following statements is very confusing and I am not able to clearly understand what authors wants to state:
- a) Page # 3 line# 62 – 64. Does environmental factors regulate genetic factors contributing to ID/ASD? If so, example supported by literature.
 - b) Page #4 line 88 – 89. What is metanalysis? There is no connection with statements mentioned in the paragraph.
 - c) It seems to me line # 91 – 99 and line #105 – 111 on the same page is out of place and needs to be restated.
 - d) Line # 120 – 123 in Page # 5 and line #170 – 173 in Page # 6 are very confusing and complex sentences.
 - e) Authors should mention visual cortex area V1 instead of just V1 for easy understanding of readers outside the field.
 - f) Line # 194 – 197 in page # 7, line # 289 – 290 Page # 10, line # 460 – 462 in Page # 16, line # 505 – 507 in Page # 17, line # 761 – 763 Page # 26 and there are similar example of confusing statements.
- 8) There are some over simplified statements. For example - “Therefore, as an adult..” line # 156 in Page 6.
- 9) Line # 210, page 8 “GABA to open critical” – Does GABA act as switch for opening / closing of critical period of plasticity or it is important for extending critical period of plasticity.
- 10) Authors should describe common strategies to make animal models, shortcoming and advantages of each of these models.
- 11) Does SYNGAP regulate mGluR LTD that is independent of FMRP-mediated protein synthesis? If so, why Syngap1 mRNA is associated with FMRP?
- 12) SYNGAP regulates spine morphology. Although, there is a mention of SYNGAP-mediated control cytoskeleton rearrangements in Figure but it is not well described in the text.
- 13) What are these contradictory evidences mentioned in line # 465 in Page 16?
- 14) What is I/O ratio?
- 15) It is not clear what is the link between mIPSC and mGluR LTD, see line # 810 – 811 , page 27.

Decision letter (RSOB-18-0265.R0)

14-Feb-2019

Dear Dr Clement,

We are writing to inform you that the Editor has reached a decision on your manuscript RSOB-18-0265 entitled "Understanding Intellectual Disability and Autism spectrum Disorder from common mouse models: synapses to behaviour", submitted to Open Biology.

As you will see from the reviewers' comments below, there are a number of criticisms that prevent us from accepting your manuscript at this stage. The reviewers suggest, however, that a revised version could be acceptable, if you are able to address their concerns. If you think that you can deal satisfactorily with the reviewer's suggestions, we would be pleased to consider a revised manuscript.

The revision will be re-reviewed, where possible, by the original referees. As such, please submit the revised version of your manuscript within six weeks. If you do not think you will be able to meet this date please let us know immediately.

When submitting your revised manuscript, please respond to the comments made by the referee(s) and upload a file "Response to Referees" in "Section 6 - File Upload". You can use this to document any changes you make to the original manuscript. In order to expedite the processing of the revised manuscript, please be as specific as possible in your response to the referee(s).

Please see our detailed instructions for revision requirements
<https://royalsociety.org/journals/authors/author-guidelines/>

Sincerely,

The Open Biology Team
mailto: openbiology@royalsociety.org

Editor's Comments to Author(s):
Please attend to all of the referees' comments.

Referee: 1

Comments to the Author(s)

This review article assesses current mouse models of ID and Autism by a junior investigator who has made contributions to the field. Overall the topic is of interest to a broad swath of neuroscientists interested in autism and synaptic mechanisms. However the manuscript needs significant editing to make it easily digestible and informative to both those in and outside of the discipline. Minor concerns are about some of the grammatical issues and awkward sentence structure which would benefit from some editorial input. But more importantly the manuscript is too diffuse without a central unifying concept. It is not clear whether the authors wish to find some convergence across the models they discuss and in fact it remains unclear why these are the

focus of the article. The authors need to articulate a critical central theme and discuss the relevant models to highlight these.

Moreover the current version reads like a laundry list of every study done in these mouse models and does not critically assess or summarize key findings. This makes the manuscript a little tedious to follow and does not put forward a more thoughtful discussion of the current state of the field. For instance many behavioral tests carried out in the *Fmr1* ko mice are listed and then it is stated that other studies cannot replicate these. The reader is left wondering what is the important finding in these mice. Some further assessment and conclusions are warranted in each case. This would also allow the authors to reduce the length of the manuscript which is currently overly lengthy.

The authors should also carefully consider what they want to discuss in the introduction. There is a lengthy discussion of critical periods and then this is not mentioned later in the context of mice. The thematic goals need to be more strongly tied together so that the reader can follow the line of thought and what are the important findings and outstanding gaps when trying to relate mouse studies to the human phenotype. For instance a discussion of critical periods and learning and memory don't really make much sense here.

Overall a major revision is required to improve this manuscript for publication.

Referee: 2

Comments to the Author(s)

The review "Understanding Intellectual Disability and Autism Spectrum Disorder from common mouse models: synapses to behaviour" by Verma et. al. highlighted recent advancement, upcoming challenges and pressing questions in the field. The review provided thorough and adequate information related to current understanding of ID and ASD. I strongly believe that the review will attract necessary attention in the field. However, the review need a major revision related to presentation of existing knowledge, English language and figures. The manuscript is good fit for publication in *Open Biology* with revision. My concerns are as follows:

- 1) The review needs to be threaded together for better appreciation of current knowledge in the field. Authors need to connect physiological relevance (electrophysiology data, spine defects and behavioural deficits) of the data with mechanistic details. In current version, these are presented in isolation.
- 2) Metaplasticity and contribution of glial cells in ID and ASD should be presented along with the section describing ID or ASD.
- 3) Conclusion section should highlight specific challenges, technical limitations and pressing questions in each of this field. It should provide information about future direction of research in each of these disorders.
- 4) I noticed authors often used colloquial terminology all throughout the manuscript. Few example - line # 46, 47, 55 in Page 2, line # 114 in Page 4, line #121 in Page 5, line 248 in page # 9, Page # 5 line 128 ..
- 5) Authors should specify specific brain area / cell type where differential expression is reported. Please see line # 55 in Page # 2 "differences in the tissue levellocalisation.." which tissue? Line # 58, page # 3 "SYNGAP1 is localized to the brain areas" which brain area?
- 6) Authors should mention how astrocyte maturation influence excitatory synapse development and why this process is linked to ID.

7) Following statements is very confusing and I am not able to clearly understand what authors wants to state:

- a) Page # 3 line# 62 – 64. Does environmental factors regulate genetic factors contributing to ID/ASD? If so, example supported by literature.
- b) Page #4 line 88 – 89. What is metanalysis? There is no connection with statements mentioned in the paragraph.
- c) It seems to me line # 91 – 99 and line #105 – 111 on the same page is out of place and needs to be restated.
- d) Line # 120 – 123 in Page # 5 and line #170 – 173 in Page # 6 are very confusing and complex sentences.
- e) Authors should mention visual cortex area V1 instead of just V1 for easy understanding of readers outside the field.
- f) Line # 194 – 197 in page # 7, line # 289 – 290 Page # 10, line # 460 – 462 in Page # 16, line # 505 – 507 in Page # 17, line # 761 – 763 Page # 26 and there are similar example of confusing statements.

8) There are some over simplified statements. For example - “Therefore, as an adult..” line # 156 in Page 6.

9) Line # 210, page 8 “GABA to open critical” – Does GABA act as switch for opening / closing of critical period of plasticity or it is important for extending critical period of plasticity.

10) Authors should describe common strategies to make animal models, shortcoming and advantages of each of these models.

11) Does SYNGAP regulate mGluR LTD that is independent of FMRP-mediated protein synthesis? If so, why Syngap1 mRNA is associated with FMRP?

12) SYNGAP regulates spine morphology. Although, there is a mention of SYNGAP-mediated control cytoskeleton rearrangements in Figure but it is not well described in the text.

13) What are these contradictory evidences mentioned in line # 465 in Page 16?

14) What is I/O ratio?

15) It is not clear what is the link between mIPSC and mGluR LTD, see line # 810 – 811 , page 27.

Author's Response to Decision Letter for (RSOB-180265.R0)

See Appendix A.

RSOB-18-0265.R1 (Revision)

Review form: Reviewer 1

Recommendation

Accept with minor revision (please list in comments)

Are each of the following suitable for general readers?

- a) **Title**
Yes
- b) **Summary**
Yes
- c) **Introduction**
Yes

Is the length of the paper justified?

Yes

Should the paper be seen by a specialist statistical reviewer?

No

Is it clear how to make all supporting data available?

Not Applicable

Is the supplementary material necessary; and if so is it adequate and clear?

Not Applicable

Do you have any ethical concerns with this paper?

No

Comments to the Author

Authors have taken comments very seriously and the revised manuscript demonstrate significant improvement. However, the English needs to be modified all through out. Authors may consider taking help from native English speaking individual.

Decision letter (RSOB-18-0265.R1)

08-May-2019

Dear Dr Clement

We are pleased to inform you that your manuscript RSOB-18-0265.R1 entitled "Understanding Intellectual Disability and Autism spectrum Disorder from common mouse models: synapses to behaviour" has been accepted by the Editor for publication in Open Biology. The reviewer(s) have recommended publication, but also suggest some minor revisions to your manuscript. Therefore, we invite you to respond to the reviewer(s)' comments and revise your manuscript.

Please submit the revised version of your manuscript within 7 days. If you do not think you will be able to meet this date please let us know immediately and we can extend this deadline for you.

- 1) A text file of the manuscript (doc, txt, rtf or tex), including the references, tables (including captions) and figure captions. Please remove any tracked changes from the text before submission. PDF files are not an accepted format for the "Main Document".
- 2) A separate electronic file of each figure (tiff, EPS or print-quality PDF preferred). The format should be produced directly from original creation package, or original software format. Please note that PowerPoint files are not accepted.
- 3) Electronic supplementary material: this should be contained in a separate file from the main text and meet our ESM criteria (see <http://royalsocietypublishing.org/instructions-authors#question5>). All supplementary materials accompanying an accepted article will be treated as in their final form. They will be published alongside the paper on the journal website and posted on the online figshare repository. Files on figshare will be made available approximately one week before the accompanying article so that the supplementary material can be attributed a unique DOI.

Online supplementary material will also carry the title and description provided during submission, so please ensure these are accurate and informative. Note that the Royal Society will not edit or typeset supplementary material and it will be hosted as provided. Please ensure that the supplementary material includes the paper details (authors, title, journal name, article DOI). Your article DOI will be 10.1098/rsob.2016[last 4 digits of e.g. 10.1098/rsob.20160049].

- 4) A media summary: a short non-technical summary (up to 100 words) of the key findings/importance of your manuscript. Please try to write in simple English, avoid jargon, explain the importance of the topic, outline the main implications and describe why this topic is newsworthy.

Images

Data-Sharing

It is a condition of publication that data supporting your paper are made available. Data should be made available either in the electronic supplementary material or through an appropriate repository. Details of how to access data should be included in your paper. Please see <http://royalsocietypublishing.org/site/authors/policy.xhtml#question6> for more details.

Data accessibility section

Sincerely,

The Open Biology Team
<mailto:openbiology@royalsociety.org>

Reviewer(s)' Comments to Author:

Referee:

Comments to the Author(s)

Authors have taken comments very seriously and the revised manuscript demonstrate significant improvement. However, the English needs to be modified all through out. Authors may consider taking help from native English speaking individual.

Decision letter (RSOB-18-0265.R2)

16-May-2019

Dear Dr Clement,

We are pleased to inform you that your manuscript entitled "Understanding Intellectual Disability and Autism spectrum Disorder from common mouse models: synapses to behaviour" has been accepted by the Editor for publication in Open Biology.

Sincerely,

The Open Biology Team
mailto: openbiology@royalsociety.org

Appendix A

Dear Editor and Reviewer's,

We thank both the reviewers for their critical comments and suggestions and for a thorough review of the manuscript. We have taken all the suggestions and comments into consideration, and all the authors' have worked on these comments and suggestions to incorporate it in the revised manuscript. We have requested our native English speaker friends to check for grammar or spelling errors. We have also used Grammarly software to check for spelling errors, appropriate sentence formation, and for grammatical errors.

Response to Reviewer's:

Referee#1

- This review article assesses current mouse models of ID and Autism by a junior investigator who has made contributions to the field. Overall the topic is of interest to a broad swath of neuroscientists interested in autism and synaptic mechanisms. However the manuscript needs significant editing to make it easily digestible and informative to both those in and outside of the discipline.

We thank the reviewer for the all comments to improve the content of the review. As per the suggestions of the reviewer, we have made considerable changes in the manuscript to improve clarity for both neuroscience as well as non-neuroscience readers. We wanted to discuss different aspects such as molecular, biochemical, and neurophysiological of a mutation which makes this review long but informative. When we wrote this view, we discussed with graduate students regarding what informations are lacking when they read a given review, and we wanted to incorporate those informations in order to bring all available informations to our readers, mainly graduate students and post-docs, and non-neuroscientists.

- Minor concerns are about some of the grammatical issues and awkward sentence structure which would benefit from some editorial input.

We thank the reviewer for the comments. We have performed a thorough check for grammatical errors in the revised manuscript. Also, we had used Grammarly software to correct any grammatical errors and make sentence structure appropriate.

- But more importantly the manuscript is too diffuse without a central unifying concept. It is not clear whether the authors wish to find some convergence across the models they discuss and in fact it remains unclear why these are the focus of the article. The authors need to articulate a critical central theme and discuss the relevant models to highlight these.

We thank the reviewer for the critical comments on the manuscript. Our idea at the time of writing this review, as the title suggests, was to summarise the animal models, particularly mouse models, available to study different monogenic mutations identified in patients with ASD/ID. However, it is evident from studies that not all animal models manifest same degree of phenotype with respect to a particular disease. Hence, it is really important to identify and elaborate on a central theme, as the reviewer has suggested in the earlier comment. To clarify, we have introduced the critical period of plasticity as one of the subsections within each gene mutations in the revised manuscript. Recent studies from the works of the corresponding author and other labs has emphasised on the fact that alteration in critical period of development is one of the major contributors to pathophysiology in ASD/ID. Therefore, we believe that it is essential to highlight critical period plasticity in the available disease models alongside with other aspects.

- Moreover the current version reads like a laundry list of every study done in these mouse models and does not critically assess or summarize key findings. This makes the manuscript a little tedious to follow and does not put forward a more thoughtful discussion of the current state of the field.

We are thankful to the reviewer for the critical comments on the manuscript. We apologise for not highlighting the critical central theme in the previous version of the manuscript, even though we have done a comprehensive literature review on almost all studies done so far. We agree, such thorough literature review without a central theme is tough to follow. Hence, as per the suggestions we have discussed the current state of the field with respect to different animal models available, as well as potential future models could be implemented, in the Discussion section of the revised review.

- For instance many behavioral tests carried out in the Fmr1 ko mice are listed and then it is stated that other studies cannot replicate these. The reader is left wondering what is the important finding in these mice. Some further assessment and conclusions are warranted in each case. This would also allow the authors to reduce the length of the manuscript which is currently overly lengthy.

We thank the reviewer for the comments to improve our manuscript. We agree that a few mouse models failed to show phenotypes similar to human patients, particularly in response to drug. As we had discussed in the animal model section on the criteria to choose a good mouse model to study a human disease/disorder, the animal model must fulfil the criteria. We wanted to emphasise how certain models do not fulfil these criteria to our readers, and the importance of choosing a good animal model. Therefore, our review could provide clarity on choosing a good animal model amongst many available animal models.

- The authors should also carefully consider what they want to discuss in the introduction. There is a lengthy discussion of critical periods and then this is not mentioned later in the context of mice.

We thank the reviewer for the comments. We apologise for the lack of clarity in the manuscript. As per the suggestion of the reviewer, we have incorporated the critical period of plasticity in the context of each mouse model.

- The thematic goals need to be more strongly tied together so that the reader can follow the line of thought and what are the important findings and outstanding gaps when trying to relate mouse studies to the human phenotype. For instance a discussion of critical periods and learning and memory don't really make much sense here.

We thank the reviewer for the critical comments. We agree that discussion on critical period of plasticity in introduction does not have an impact, unless critical period of plasticity for all the mutations is available for readers. In the revised manuscript, we have added critical period of plasticity for all mutations, as suggested by the reviewer.

We have also highlighted the gaps in the available mouse models in the Discussion section. In addition, we have discussed the importance of using human derived iPSCs to study ID/ASD as well as to identify drugs specific for a mutation, if possible.

- Overall a major revision is required to improve this manuscript for publication.

We thank the reviewer for all the comments and suggestions to improve the manuscript. We have incorporated all the changes as suggested by the reviewer.

Referee#2

- The review “Understanding Intellectual Disability and Autism Spectrum Disorder from common mouse models: synapses to behaviour” by Verma et. al. highlighted recent advancement, upcoming challenges and pressing questions in the field. The review provided thorough and adequate information related to current understanding of ID and ASD.

We thank the reviewer for all the comments and suggestions.

- I strongly believe that the review will attract necessary attention in the field. However, the review need a major revision related to presentation of existing knowledge, English language and figures. The manuscript is good fit for publication in Open Biology with revision.

We thank the reviewer for the comments. We have performed a thorough check for grammatical and spelling errors and modified the language to a formal English in the revised manuscript. Also, we had used Grammarly software to correct any grammatical errors and make sentence structure appropriate.

- My concerns are as follows:
 - 1) The review needs to be threaded together for better appreciation of current knowledge in the field. Authors need to connect physiological relevance (electrophysiology data, spine defects and behavioural deficits) of the data with mechanistic details. In current version, these are presented in isolation.

We thank the reviewer for the suggestions and comments on our manuscript. As the reviewer has suggested, we have linked the data presented in the section of different mutations to its physiology and behaviour in the revised manuscript. Thus, we believe it will bring clarity to the readers to correlate electrophysiological, imaging and behaviour data. In addition, we have correlated synaptic deficit data with behavioural phenotype which will further bring clarity to the readers.

2) Metaplasticity and contribution of glial cells in ID and ASD should be presented along with the section describing ID or ASD.

We thank the reviewer for the suggestion. The studies involving metaplasticity and glial cells in ID/ASD are limited, particularly for different mutations. As of now, there is a lack of data available for metaplasticity in the literature for any of the ID/ASD mentioned in the manuscript. Similar, studies on the function of astrocytes in pathophysiology in ID/ASD is available only for Fragile X

syndrome and not for any other mutations mentioned in this manuscript. However, recently, a few labs have started research on the function of astrocytes in ID/ASD, but those labs have only preliminary evidences. Therefore, we have written this section separately, instead of writing it for each mutation, so that the readers can understand the lack of information related to metaplasticity and astrocytes in these mutations.

3) Conclusion section should highlight specific challenges, technical limitations and pressing questions in each of this field. It should provide information about future direction of research in each of these disorders.

We thank the reviewer for the crucial suggestions and comments on our manuscript. We have incorporated all the suggestions in the revised manuscript.

4) I noticed authors often used colloquial terminology all throughout the manuscript. Few example – line # 46, 47, 55 in Page 2, line # 114 in Page 4, line #121 in Page 5, line 248 in page # 9, Page # 5 line 128 .

We apologise for these errors. We thank you for the detailed comments. We have now made appropriate corrections in the revised manuscript.

Previous version: Line 46, 47 in Page 2; ‘symptoms which distinguishes patients with ID is their inability to take care of themselves for normal day to day activities’

Revised version: The colloquial sentence has been removed and the self-care aspect has been included in the line 43

Previous version: Line 55 in page 2; ‘While the distribution of certain proteins like FMRP is widespread, the expression for other proteins like SYNGAP1 is localised to the brain areas,’

Revised version: Line 53 in page 2; ‘While the distribution of certain proteins like FMRP is widespread, the SYNGAP1 protein is expressed only in the brain, not in any other organ.’

Previous version: Line 114 in page 4; ‘However, they are classifiable based on the timing of causation (concerning birth), the nature of the causal factor (genetic, environmental or unknown) or a combination of both.’

Revised version: Line 118 in page 5; ‘The causes of ID and ASD are multifarious and involve a range of genetic to environmental factors.’

Previous version: Line 121 in page 5; ‘Environmental input during this period plays an instrumental role in shaping the development and function of the brain’

Revised version: Line 123-128 in page 5; ‘Environmental stress factors like poor nutrition, hygiene, infection, familial instability, and socio-economic causes, may affect brain development contributing to ID/ASD. For example, it is now believed that the oxidative stress as a result of environmental stressors like heavy metals affects sulphur metabolism which leads to alteration of epigenetic mechanisms of gene expression. Hence, the complex interplay of genetic and environmental factors plays a crucial role in the pathology of ASD/ID’.

Previous version: Line 128 in page 5; ‘Non- inherited or acquired genetic factors are a majorly studied cause of ID.’

Revised version: The colloquial line has been removed

Previous version: Line 228 in page 9; ‘However, it is not possible to mimic all aspects of a disorder or disease in any one animal model.’

Revised version: Line 244-246 in page 9; ‘Although it is not possible to mimic all aspects of a disorder or disease in any one animal model, a good animal model should be able to replicate the clinical hallmarks of the disease with the paramount degree of robustness.’

5) Authors should specify specific brain area / cell type where differential expression is reported. Please see line # 55 in Page # 2 “differences in the tissue level localisation..” which tissue? Line # 58, page # 3 “SYNGAP1 is localized to the brain areas” which brain area?

We thank the reviewer for bringing this to our notice. For clarity, we have incorporated these suggestions.

Previous version: Line 55-58; ‘Differences in the tissue level localisation of the proteins may also be responsible for the severity of symptoms associated with a mutation in the genes coding for these proteins. While the distribution of certain proteins like FMRP is widespread, the expression for other proteins like SYNGAP1 is localised to the brain areas, particularly in the excitatory postsynapses,’

Revised version: Line 53-54; ‘While the distribution of certain proteins like FMRP is widespread, the SYNGAP1 protein is expressed only in the brain, not in any other organ’

6) Authors should mention how astrocyte maturation influence excitatory synapse development and why this process is linked to ID.

We thank the reviewer for the suggestion. We have briefly discussed how astrocyte maturation influences excitatory synapse development in the revised manuscript. We have included that the secretion of molecules such as glypicans by astrocytes which help in the conversion of silent to functional synapses, thereby, facilitating the insertion of AMPA receptors. On this basis one can speculate that any alteration in this process might alter the critical period of plasticity which in turn might lead to ID/ASD, although, there is not direct evidence as of now.

Previous version: Line 77-78; 'It has been observed that the timing of the astrocyte maturation coincides well with the formation of the excitatory synapses in the brain.'

Revised version: Line 74-77; 'It has been observed that the timing of the astrocyte maturation coincides well with the formation of the excitatory synapses in the brain. Secretion of molecules such as glypicans by astrocytes helps in the conversion of silent to functional synapses, thereby, facilitating the insertion of AMPA receptors.'

7) Following statements is very confusing and I am not able to clearly understand what authors wants to state:

a) Page # 3 line# 62 – 64. Does environmental factors regulate genetic factors contributing to ID/ASD? If so, example supported by literature.

We thank the reviewer for the comments. We have incorporated these changes and discussed briefly on the environmental factors regulating genetic factors contributing to the ID/ASD.

Previous version: Line 62 – 64: Amongst the environmental factors which can lead to ID and ASD include the use of certain drugs during pregnancy like valproate, alcohol, infections, exposure to heavy metals like lead and mercury, and malnutrition. However, many cases of ID are known to be idiopathic

Revised version: Line 58-62: 'Amongst the environmental factors which can lead to ID and ASD include the use of certain drugs during pregnancy like valproate, alcohol, infections, exposure to heavy metals like lead and mercury, and malnutrition. Though a complex interplay of environmental and genetic factors is known to play a role in the pathophysiology of ID/ ASD, most cases are idiopathic.'

Additionally, we have incorporated the suggestions in the section, causes of intellectual disability, line 125-128 ‘For example, it is now believed that the oxidative stress as a result of environmental stressors like heavy metals affects sulphur metabolism which leads to alteration of epigenetic mechanisms of gene expression. Hence, the complex interplay of genetic and environmental factors plays a crucial role in the pathology of ASD/ID.’

b) Page #4 line 88 – 89. What is metanalysis? There is no connection with statements mentioned in the paragraph.

We thank the reviewer for the comments. We have modified this statement to bring clarity to our readers.

Previous version: Line 88-89; ‘Metanalysis studies have highlighted the procedural limitations and underreporting of the cases from several parts of the world’.

Revised version: Line 90-92; ‘Metanalysis studies, which make use of the statistical procedures to analyse data from already existing reports, have highlighted the procedural limitations and underreporting of the cases from several parts of the world’.

c) It seems to me line # 91 – 99 and line #105 – 111 on the same page is out of place and needs to be restated.

We thank the reviewer for the comments. We apologise for these errors. We have edited these sentences.

Previous version: Line 90-99; ‘Additionally, the prevalence was higher in the low- and middle-income countries which could attribute to the lack of fundamental diagnostic and management resources in these geographical locations. An exhaustive study done in children from birth till 12 years of age had found that the prevalence of NDD is as high as 1 in 6 children in the United States of America. Moreover, in the same study, the incidence was higher amongst the males in comparison to the females However, according to the WHO’s recent report, about 1 in every 160 individuals suffer from ASD/ID. The numbers are remarkably alarming as it is only expected to become worse in the absence of any effective therapeutic strategy. With more number of laboratories now working on NDDs, insights into new possible therapeutic targets and their mechanisms may aid to find mitigation strategies in the future.’

Revised version: Line 93-98; ‘However, current studies have shown the prevalence to be higher in the low- and middle-income countries which could attribute to the lack of fundamental diagnostic and management resources in these geographical locations. Considering NDDs requires early diagnosis, an exhaustive study done in children from birth till 12 years of age had found that the prevalence to be as high as 1 in 6 children in the United States of America. Moreover, in the same study, the incidence was higher amongst the males in comparison to the females.’

Previous version: Line 105-111; ‘The involvement of synaptic function and plasticity in ID and ASD is well characterised. With the advancement in technology to measure neuronal activity (electrophysiology and deep-brain imaging) and understanding of the mechanisms which control synaptic plasticity may aid in further expanding our knowledge to decipher the pathophysiology of ID and ASD. For example, metaplasticity, which is one such checkpoint on the synaptic plasticity, might be altered in the ID and ASD. However, in the absence of any substantial experimental evidence, it remains to be established if it plays any significant role in the pathophysiology of ID and ASD.’

Revised version: Line 109-116; ‘Although there are several unexplored mechanisms associated with ID/ASD, the involvement of synaptic function and plasticity in ID and ASD is well characterised. With the advancement in technology to measure neuronal activity (electrophysiology and deep-brain imaging) and understanding of the mechanisms which modulates synaptic plasticity may aid in further expanding our knowledge to decipher the pathophysiology of ID and ASD. In this review, we will discuss what are the implications of monogenic mutations on the physiological, molecular and biochemical, and morphological aspects of neuronal and neuronal development using different mouse models from the studies carried out in the past few decades.’

d) Line # 120 – 123 in Page # 5 and line #170 – 173 in Page # 6 are very confusing and complex sentences.

We thank the reviewer for the comments. We apologise for these errors. We have corrected these errors in the revised manuscript.

Previous version: Line 120-123 in Page #5; Environmental input during this period plays an instrumental role in shaping the development and function of the brain. Thus, stress factors of environmental origin such as poor nutrition, hygiene, infection, familial instability, most likely having socio-economic roots, can cause improper brain development leading to ID/ASD.’

Revised version: Line 123-128 in Page #5; Environmental stress factors like poor nutrition, hygiene, infection, familial instability, and socio-economic causes, may affect brain development contributing to ID/ASD. For example, it is now believed that the oxidative stress as a result of environmental stressors like heavy metals affects sulphur metabolism which leads to alteration of epigenetic mechanisms of gene expression. Hence, the complex interplay of genetic and environmental factors plays a crucial role in the pathology of ASD/ID.

Previous version: line #170 – 173 in Page # 6; Similarly, onset of hearing triggers critical period of development of auditory cortex in humans. In the recent decade or so, several labs have used visual and thalamocortical regions of the brain as a model to understand the impact of occlusion of sensory input impairs formation, maturation, and elimination of neuronal connections.

Revised version: line #166-170 in page #6; Similarly, the onset of hearing triggers critical period of development of auditory cortex in humans. Perturbation in neuronal connections of these regions before the end of the critical period might permanently compromise their function. In the recent decade or so, several labs have used visual, auditory, and thalamocortical regions of the brain as a model to understand the importance of critical period in plasticity and development of an individual.

e) Authors should mention visual cortex area V1 instead of just V1 for easy understanding of readers outside the field.

We thank the reviewer for the comments. We have incorporated these changes

f) Line # 194 – 197 in page # 7, line # 289 – 290 Page # 10, line # 460 – 462 in Page # 16, line # 505 – 507 in Page # 17, line # 761 – 763 Page # 26 and there are similar example of confusing statements.

We thank the reviewer for the critical comments on our manuscript. We have rephrased the sentences to bring more clarity.

Previous version: Line 289-290 in page 10; ‘These mice showed a significant impairment in the working memory but performed comparably to the wild-type animals in the reference memory tasks’

Revised version: Line 280-282 in page 10; ‘Syngap1+/- mice showed a significant decline in the working memory, though, the performance was comparable to the wild-type animals in the reference memory tasks suggesting only certain memories are impaired.’

Previous version: Line 460-462 in page 16; ‘However, it cannot be conclusively said that the *Fmr1*-KO mice model successfully recapitulates the human behavioural phenotypes of the fragile x syndrome as multiple studies have reported contradictory evidence to all the tests mentioned above’.

Revised version: Line 495-497 in page 17; ‘However, caution should be shown to determine a suitable mouse model as it should fulfil the criteria discussed in the animal model section earlier’.

Previous version: Line 505-507; ‘In general, reduced inhibition and increased excitation levels were observed in these mice, corroborative with the increased susceptibility to seizures, that might result due to imbalanced levels of excitatory, inhibitory, and neuromodulatory signalling in the brain’.

Revised version: Line 536-537; ‘In general, reduced inhibition and increased excitation levels were observed in these mice, corroborative with the increased susceptibility to seizures and epileptogenesis’.

Previous version: Line # 194 – 197 in page # 7; For example, based on these studies, NMDARs are considered as one of the molecular determinants of the critical period of plasticity as NMDAR-mediated synaptic transmission are developmentally regulated and their expression could be modified by neuronal activity [58, 59]. Depending on the expression of a subunit of NMDAR, NR2A or NR2B, the current mediated from these subunits varies during developmental stages that suggest closure of the critical period, thereby, signalling a change in the processing of information.

Revised version: Line # 190 – 193 in page # 7; For example, based on these studies, NMDARs are considered as one of the molecular determinants of the critical period of plasticity as NMDAR-mediated synaptic transmission are developmentally regulated, and their expression could be modified by neuronal activity.

Previous version: line # 761 – 763 Page # 26; Since ID/ASD is one of the neurodevelopmental disorders, understanding the role of these proteins in synaptic function during early stages of development might give insights into the role of NLGNs in critical period of development regulation via NLGNs. For studying the correlation of NLGNs on the critical period of development, monocular and binocular deprivation (MD, BD) in mice was performed.

Revised version: Above sentences has been removed as they were discussed later in the section.

8) There are some over simplified statements. For example - “Therefore, as an adult..” line # 156 in Page 6.

We thank the reviewer for the critical comments on our manuscript. We have rephrased the sentences as suggested by the reviewer.

Previous version: 156 in Page 6; Therefore, as an adult, it is slightly difficult to learn new things at faster pace than in childhood.

Revised version: 154-155 in Page 6; For example, a child may learn a new language in a few months to a year, but, it might take years for an adult to learn the same language.

9) Line # 210, page 8 “GABA to open critical” – Does GABA act as switch for opening / closing of critical period of plasticity or it is important for extending critical period of plasticity.

We thank the reviewer for the critical comments on our manuscript. GABA functions via two Cl⁻ cotransporters, NKCC1 and KCC2 (whose expressions are developmentally regulated), present in the brain. Switch-over of NKCC1 to KCC2 marks the end of the critical period of development and plasticity. Therefore, indirectly, GABA serves as a switch for opening / closing of critical period of plasticity. It could be used as an essential therapeutic target for extending the critical period of plasticity in the neurodevelopmental disorders.

10) Authors should describe common strategies to make animal models, shortcoming and advantages of each of these models.

We thank the reviewer for the comments. This is incorporated as per the reviewer's suggestions.

Revised version: Line 1071-1084; ‘Preclinical studies in this regard can warrant some useful clues for the translational success of small molecules.....Such combinatorial study can fulfil the existing gap in our knowledge about ID/ASD and show the way for future therapeutic strategies’.

11) Does SYNGAP regulate mGluR LTD that is independent of FMRP-mediated protein synthesis? If so, why *Syngap1* mRNA is associated with FMRP?

We thank the reviewer for the comments. In *syngap1* heterozygous condition, mGluR-LTD was shown to be increased (We have described it in the manuscript). However, the mechanism behind the same is not known, in the context of *Syngap1* heterozygous. Whereas, FMRP does regulate

Syngap1 mRNA translation by associating with it (accepted in Front. In Mol. Neuroscience). These observations suggest that there could be a crosstalk between FMRP and SYNGAP1.

We apologise for the lack of clarity in the manuscript. We have rephrased those sentences to make it clear to the readers.

Previous version: Line 373-375; ‘Our unpublished data and their results showed an increase in LTD in the *Syngap1*^{+/-} mice which were independent of the protein synthesis, but, the results were similar to *Fmr1*^{-/-} suggesting a possible regulation of *Syngap1* mRNA by FMRP’.

Revised version: Line 375-382; ‘To study the role mGluR-mediated LTD, Barnes *et al.* stimulated Group I mGluRs with Dihydroxyphenylglycine (DHPG) in the hippocampus and showed mGluR-LTD was significantly increased and independent of protein synthesis in *Syngap1*^{+/-} mice at PND 25-32..... suggests a presence of convergence in the biochemical signalling pathway downstream of mGluR- and NMDAR- mediated signalling proteins, although it needs to be explored’.

12) SYNGAP regulates spine morphology. Although, there is a mention of SYNGAP-mediated control cytoskeleton rearrangements in Figure but it is not well described in the text.

We thank the reviewer for the comments. As per the suggestion, we have incorporated SYNGAP1’s role in cytoskeleton rearrangements in the revised version.

Revised version: Line 393-396; ‘Additionally, the activity of proteins like p21-Activated Kinase, RAC, and p-Cofilin, which are regulated by the SYNGAP1, were also elevated in the *SYNGAP1*^{+/-} mice under the basal conditions. Thus, SYNGAP1 regulates spine morphology and function by modulating cytoskeletal dynamics.’

13) What are these contradictory evidences mentioned in line # 465 in Page 16?

We thank the reviewer for the comments. As per the suggestion, we have reframed the sentence to make it reader friendly and one unified viewpoint has been highlighted.

14) What is I/O ratio?

We thank the reviewer for the comments. I/O ratio is input/output ratio. It’s the measure of basal synaptic transmission of the neurons which means there should be a corresponding output to every input provided to the neurons.

15) It is not clear what is the link between mIPSC and mGluR LTD, see line # 810 – 811 , page 27.

We thank the reviewer for the comments. Patch clamp recordings from the cerebellum of *Nlgn3* KO mice has revealed increased mIPSC (increased number of functional GABA receptors) and decreased mEPSC (decreased number of functional AMPA receptors) in the cerebellum, suggesting increased inhibitory activity as a number of functional GABAR are more as compared to excitatory activity. Additionally, since there was decreased mEPSC (decreased number of functional AMPA receptors in the postsynapse, and LTD, which is the correlate for endocytosis of AMPAR from the postsynapse), it has been observed impairment of mGluR-mediated long-term depression (LTD). We have incorporated revised these changes in the manuscript.

Previous version: line # 810 – 811 , page 27; Patch clamp recordings from the hippocampus, somatosensory cortex, and cerebellum of *Nlgn3* KO mice has revealed increased mIPSC and decreased mEPSC in the hippocampus, and decreased mEPSC and impaired mGluR-mediated long-term depression (LTD) in the cerebellum.

Revised version: line # 883 – 890 , page 30; Patch clamp recordings from the hippocampus, somatosensory cortex, and cerebellum of *Nlgn3* KO mice has revealed increased mIPSC (increased number of functional GABA receptors) and decreased mEPSC (decreased number of functional AMPA receptors) in the hippocampus, suggesting increased inhibitory activity as a number of functional GABAR are more as compared to excitatory activity. Additionally, decreased mEPSC and impaired mGluR-mediated long-term depression (LTD) in the cerebellum was also observed, which suggests the differential function of *Nlgn3* in different regions of the brain